# Have China’s Pilot Free Trade Zones Improved Green Total Factor Productivity?

**DOI:** 10.3390/ijerph182111681

**Published:** 2021-11-07

**Authors:** Qingshan Ma, Yuanmeng Zhang, Kexin Yang, Lingyun He

**Affiliations:** 1School of Economics, Xiamen University, No. 422 Siming South Road, Xiamen 361005, China; maqingshan@stu.xmu.edu.cn; 2Economics School, Zhongnan University of Economics and Law, No. 182 Nanhu Avenue, Wuhan 430073, China; yuanmeng_zhang@foxmail.com; 3Economics and Management School, Wuhan University, No. 299 Bayi Road, Wuhan 430072, China; kexin_yang@whu.edu.cn

**Keywords:** pilot free trade zones, green total factor productivity, green development, difference-in-differences, quasi-natural experiment

## Abstract

Free trade zones (FTZ) are designated areas for promoting trade openness and investment facilitation. In China, FTZs are also regarded as “green areas” in which planning actions and institutional innovations are implemented, and there is a commitment to promoting urban green and healthy development. Given that green total factor productivity (*GTFP*) is an important measure of a city’s health and green performance, this study exploits the difference-in-differences method to explore the impact of pilot FTZs on urban *GTFP* in 280 cities in China for the period between 2005 and 2017. The results show that the green areas positively contributed to the growth of *GTFP*. Moreover, the outcome holds with robustness tests. Statistically, the positive effect emerged in cities during the first three years after introducing the initiative, with the effect disappearing afterward. It also had a strong positive impact in the central and western regions and in large and medium-sized cities, while the influence remained insignificant in the remaining areas in China. Furthermore, the paper also reveals that the promotion of foreign direct investment and industrial structure upgrading are the primary channels through which the positive relationship between pilot FTZs and *GTFP* is established.

## 1. Introduction

The Chinese economy has undergone spectacular growth in recent decades, while environmental degradation from traditional extensive economic development is becoming an extremely serious problem. Environmental pollution brings challenges to the sustainable and healthy development of cities [1]. According to “the Report on the State of the Ecology and Environment”, in 2020, 40.1% of the 337 cities at or above the prefecture level exceeded air quality standards. Severe environmental pollution endangers people’s health [2], which results in inequality [3], aggravates public finance burden [4], and hinders the process of urban renewal [5]. Therefore, the need for urban green and sustainable transition has become inevitable. In this context, the Chinese government has released a series of institutional innovations and policy experiments to control environmental pollution and improve the urban population’s health. One of these innovations is the free trade zone (FTZ) policy.

According to Teifenbrun [6] and Akbari et al. [7], an FTZ refers to an area where goods may be landed, handled, manufactured, or reconfigured, and then re-exported without the intervention of customs authorities. In the zones, multiple institutional incentives such as free movement of goods and people and preferential taxation are provided. Since the 1980s, countries such as the United States and Brazil have established FTZs, which are conductive to attract foreign direct investment (FDI), and greatly promote trade development [8,9,10]. The Chinese government has always been committed to deepening reform and opening wider to the outside world. In 1980, Shenzhen, Zhuhai, Shantou, and Xiamen were approved as special economic zones (SEZ). Then, China established the first bonded zone (BZ) in Shanghai in 1990, and successively set up BZs in Tianjin, Dalian, and other cities. In 2000, China approved 15 pilot export processing zones (EPZs) including Yantai Weihai, Hangzhou, at one time. Since then, China has continuously established some other SEZs, BZs, and EPZs. On 29 September 2013, China’s first FTZ, the Shanghai pilot FTZ, was established, which marks a new stage of China’s opening-up. After that, more pilot FTZs were built in Tianjin, Chongqing, and other cities by March 2017. 

Compared to previous policy zones like SEZs, the opening-up of the service and financial industries has been expanded in pilot FTZs for the first time. The Chinese pilot FTZs have also implemented a series of institutional innovations in addition to possessing functions similar to those of other countries. The institutional innovations mainly include: first, optimizing government functions and improving trade and investment efficiency through one-window service; second, tax holidays and tax regimes such as reduced income tax for some enterprises in pilot FTZs; third, introducing the “Negative List” management model to provide guidance and governs industry sectors in which foreign investment is prohibited or where possible restrictions may apply. 

“Green development” is a major development idea of Chinese pilot FTZs. As the concepts emphasized in the “Guiding Opinions on Strengthening Ecological Environment Protection in Pilot Free Trade Zones” issued by the Ministry of Ecology and Environment (MEE) of China, the ideas of green development should penetrate to the entire process of pilot FTZs construction, and pilot FTZs must develop a modern green service industry, green manufacturing, green supply chain, and green trade in the future. Some studies also indicated that pilot FTZs make efforts to become green areas in various ways [11]. Therefore, as green areas, have the pilot FTZs effectively promoted urban green development? This paper attempts to address this question and provide useful policy alternatives.

Numerous studies have examined the economic effects of the policy zones. The literature has indicated that Chinese SEZs, EPZs, and BZs are all conducive to attracting more investors and promoting trade development [12,13,14]. These special zones gain other benefits such as participation in global value chains and knowledge exchange [15,16]. Some researchers have indicated that FTZs also have effects on attracting FDI and promoting trade development [17]. As suggested by Jiang et al. [18], pilot FTZs can effectively combine domestic production factors with advanced international technology. Moreover, pilot FTZs greatly promote cross-country knowledge spillover and improve China’s competitive advantages in global industrial and supply chains [19,20]. Additionally, Song et al. [21] found that the policy advantages of pilot FTZs are beneficial to upgrading the organization and performance of local export-oriented enterprises.

Although few studies have attempted to estimate the impact of policy zones on urban green and healthy development, many studies have explored their impact on economic growth. Most extant studies support that SEZs, EPZs, and BZs positively contribute to innovation stimulation and economic growth [22,23]. However, additional critical findings suggest that these policies also have some dampening effects such as resource mismatches [24], widening the gap between the rich and poor [25], and tax evasion [26]. Concerning FTZs, the existing literature mainly focuses on positive effects. For instance, some studies have found that FTZs play an important role in innovation stimulation and enhancing the competitiveness of enterprises, which promotes economic growth [18,27,28,29,30]. From the perspective of Chinese pilot FTZs, Tan et al. [31] focused on the Shanghai pilot FTZ and provided evidence that there is a significant and positive effect on the growth rate of the total imports and exports of Shanghai. Based on the difference-in-differences (DID) method, Zhang et al. [32] found that pilot FTZs drive regional economic growth. Chen et al. [33] also suggested that China’s FTZ policy is effective and increases the national economic welfare.

Another emerging stream of literature emphasizes the impact of policy zones on environmental pollution. Some researchers argue that the policy zones promote environmental protection, whereas other researchers find that these zones generate high emissions of metals and other negative products that contaminate the environment [34,35]. Regarding FTZs, it is generally believed that Chinese pilot FTZs have reduced environmental pollution, since various measures such as clean production mechanisms are adopted by the government for pollution prevention and control [36,37].

Based on these two strands of literature, the important question that we focus on is whether the economic growth brought by pilot FTZs is at the expense of the environment or has green characteristics. Currently, green total factor productivity (*GTFP*) has been widely used in the research field of environmental economics to reflect the level of green development [38,39]. Previous literature has regarded total factor productivity (TFP) as an important measure of economic growth [40,41,42]. These studies conducted related research by taking only the traditional input (e.g., capital and labor) and desirable output (e.g., GDP) into account, while ignoring the green and environmental factors. With the development of society, the constraints of the environment on economic growth have become increasingly prominent. Therefore, it is not sufficient to use the traditional TFP indicator to analyze green and sustainable economic development. *GTFP* is a new productivity indicator that comprehensively incorporates resource and environmental constraints. Specifically, the new indicator adds resource consumption as an input factor to the traditional TFP analysis framework and adds environmental pollution emissions as an undesirable output to the input-output efficiency analysis [25,43,44], which can effectively reflect the level of green and healthy development [45,46].

At present, a small amount of attention has been given to the impact of pilot FTZs in other countries on regional green development [7,47,48]. However, concerning China, regarding pilot FTZs and green development, only a few researchers such as Jiang et al. [18], have conducted research on the Shanghai pilot FTZ and found that pilot FTZ is a great incentive to promote green development. However, Zhuo et al. [49] took the Guangdong pilot FTZ as the research object and concluded that this FTZ operates at the expense of the environment: for every 100 million yuan increase in the GDP, discharged wastewater and waste gas increases by 1.746 million tons and 28.016 tons, respectively.

According to the abovementioned literature, the following problems still need to be remedied. First, current studies mainly explore the impact of pilot FTZs on economic growth, but few studies have focused on the relationship between pilot FTZs and green development. Clarifying the relationship between pilot FTZs and green development is of great significance for providing policy implications to promote urban green development through institutional innovations. Second, the existing literature on pilot FTZs and green development focuses only on the case studies of a special pilot FTZ, and the conclusions remain inconsistent and controversial. Therefore, the relationship between pilot FTZs and urban green development is uncertain, and we do not know whether the effect is of universal significance.

Therefore, the present study takes advantage of the quasi-experiments provided by the policy of FTZ construction and adopts the DID method to explore the impacts of pilot FTZs on urban green development. This study contributes to the existing literature in two ways. First, our article contributes to the literature on the economic and environmental impacts of different policy zones. We discuss the impact of the ideas of green development and institutional innovation measures implemented in pilot FTZs on the relationship between free trade and environmental pollution. Therefore, the research in this paper is helpful for a deeper understanding on how to exert government functions to better promote urban green development. Second, this paper takes the pilot FTZs as research objects and uses the DID and propensity score matching-DID (PSM-DID) methods to investigate the impact of pilot FTZs on urban green development, which can address the endogeneity problem, and ensure the reliability of the research conclusions.

The structure of the rest of paper is as follows. The institutional background and theoretical hypothesis are explained in the second section. The data and methodology are reported in the third section. The empirical results are proposed in the fourth section. Exploring the mediating effect of FTZ policy on *GTFP* is explored in the fifth section. Finally, conclusions are presented in the sixth section.

## 2. Institution Background and Theoretical Hypothesis

### 2.1. Institution Background

Since 1978, China has established SEZs. The zones were designed as major platforms that provided preferential policies such as tax reduction for foreign enterprises. In addition, BZs have been successfully constructed since the 1990s. Goods are exempt from duty in these zones. Further, the Chinese government set up EPZs where goods manufactured for export are exempt from tax.

Government’s peculiarities and intervention play an important role in the development of these policy zones [50,51,52]. In the Chinese context, the key experiences of SEZs, EPZs, and BZs can best be summarized as gradualism with an experimental approach; a strong commitment; and the active, pragmatic facilitation of the state [53]. First, the Chinese government provides preferential policies and broad institutional autonomy such as duty-free benefits, which greatly promoted industrial clusters. Second, strong support and proactive participation of governments are provided, especially in the areas of public goods and externalities, which ensure a stable and supportive environment for long-term development. Third, the government continuously promotes technology learning and upgrading by designing policies and foresight activities.

Under the positive impact and intervention of government, most of special zones in China, though differing in characteristics, are quite successful in FDI introduction and trade prosperity. However, there is still considerable room for China to expand the opening of its service industry and attract advanced foreign investment.

To further accelerate the pace of opening to the outside world, on 27 September 2013, the first Pilot FTZ (Shanghai) in mainland China was founded. Up to March 2017, the Chinese government has established a total of 11 pilot FTZs. The policies of pilot FTZs include not just the free entry and exit of goods, zero-tariff policies, and other traditional policies that promote trade liberalization. More importantly, the FTZ policy is China’s first policy to expand the service industry and financial opening after joining the WTO. For example, the “Negative List” management model and “pre-entry national treatment” were adopted in pilot FTZs to achieve expanding openness in the fields of the modern services industry, such as banking, insurance, capital markets, and other fields. On the premise of complying with relevant regulations, qualified foreign-funded institutions and organizations are allowed to establish companies to conduct related business in the form of joint ventures or independent forms, while enjoying various preferential policies. In addition, the convertibility of the RMB capital account, the marketization of interest rates in the financial market, the reform of foreign exchange management, the opening of financial institutions, and the construction of international block trading platforms are all piloted in pilot FTZs. Compared with other special policy zones, FTZs are multifunctional areas that gather multiple innovational advantages, as shown in Table 1.

Different from the FTZs of other countries, pilot FTZs in China have implemented a series of institutional innovations to improve trade efficiency, investment efficiency, and administrative efficiency. Institutional innovation is reflected in the simplified customs clearance procedures for commodities and higher administrative efficiency. For example, pilot FTZs apply the “first entering and then declaring” process and simplify customs clearance documents, and so on. In addition, pilot FTZs have vigorously promoted the reform of e-government, improved the administrative examination and approval efficiency of the enterprises in the zone. Moreover, high-tech enterprises, financial service enterprises, cultural and creative enterprises, and many other industries in the zones can enjoy lower income tax rates. Overseas high-tech talents in pilot FTZs can also enjoy personal income tax relief. These measures have effectively promoted the development of emerging industries and economic growth in pilot FTZs.

FTZs are also set as demonstration zones for protecting the environment and achieving high-quality development [54]. “Opinions of the Ministry of Commerce on Supporting the Innovative Development of Pilot Free Trade Zones 2015” proposes to continuously explore the establishment of a green supply chain management system and encourage the development of green trade. “The Guiding Opinions on Strengthening the Ecological Environment Protection and Promoting High-quality Development in Free Trade Zones” released by the MEE of China highlighted the need to explore institutional innovations related to environmental protection in these green areas. Currently, all planned actions of the 21 existing pilot FTZs include provisions for promoting green development. For instance, “Article 50 of the Regulations of China (Shanghai) Pilot Free Trade Zone” clearly stipulates that pilot FTZs should strengthen environmental protection, improve environmental assessment classification management, and use the advantage of openness to be in line with international environmental and energy system standard certification. Shenzhen is committed to ensuring 100% coverage of green buildings, establishing a green and creative transportation system, and constructing a central cooling project in the zone to reduce pollutant emissions. Other pilot FTZs are also actively engaged in green development, for instance, the Nanjing pilot FTZ develops green finance. Figure 1 shows that the average investment in environmental governance of the provinces where pilot FTZs are located has increased from 66.11 billion yuan in 2003 to 310.44 billion yuan in 2017, and after the FTZ policy emerged, it maintains a continuous upward trend, which fully reflects the green development effects created by FTZs.

### 2.2. Hypotheses

An important influencing mechanism of pilot FTZs on urban *GTFP* is investment facilitation. Yao et al. [55], Qu et al. [56], and Tan and Yan [57] found that pilot FTZs have a strong foreign investment-induced effect. FTZs are committed to attracting the entry of foreign-funded enterprises with high productivity. The preferential policies of pilot FTZs have abolished many of the previous restrictions on the share of FDI, relaxed investment restrictions, and continued to simplify investment procedures. Both the Shanghai pilot FTZ and the Guangdong pilot FTZ have implemented preferential tax policies for high-tech enterprises and access and work convenience policies for high-tech talent. Pilot FTZs have gradually become a gathering place for high-end foreign enterprises. On the one hand, with the expansion of the scale of enterprises that gather in the pilot FTZs, different enterprises can establish connections with one another through resource sharing and the division of labor based on specialization, which effectively reduces transaction costs, and increases productivity [58]. At the same time, the knowledge spillover effect allows the spread of advanced technology and management experience in the agglomeration zone, thereby enabling the enterprises in the zone to learn and use more advanced energy-saving technologies and management methods [59]. The two effects work synergistically to continuously improve the production and environmental efficiency of the enterprises in the zones, which, in turn, has a positive impact on improving urban *GTFP*. On the other hand, the inflow of FDI squeezes the market share of domestic enterprises to a certain extent, intensifies industry competition, and creates a selection mechanism for the survival of the fittest. This phenomenon strongly motivates domestic enterprises to carry out technological innovation, thereby promoting the improvement of urban *GTFP*. Therefore, we put forward the following hypothesis:

**Hypothesis** **1** **(H1).***Pilot FTZs can increase GTFP through the promotion of FDI*.

Industrial upgrading is another influencing mechanism of pilot FTZs on urban *GTFP*. As an innovation in the trade management system, pilot FTZs play an active role in breaking down trade barriers, overcoming obstacles to factor flow, and making advanced resources flow freely on an international scale. On the one hand, the main policy goal for China to set up pilot FTZs is to accelerate the opening of the service and financial industry, which is beneficial to the upgrading of the industrial structure [60]. Upgrading of the industrial structure can not only reduce the emission of pollutants but also accelerate green development [61]. Therefore, it has a considerably positive effect on the growth in urban *GTFP*. On the other hand, the trade facilitation brought by pilot FTZs also brings new opportunities for industrial upgrading. The special policies of pilot FTZs can not only simplify international trade procedures and reduce international trade costs but also improve logistics efficiency, which effectively promotes the flow of high-tech production factors [62,63]. Advanced production factors continuously flow to high value-added industries, which can not only facilitate the development of import, export, and service trade, enhance the competitiveness of related industries, but also optimize the industrial structure and promote industrial upgrading [29,64]. In addition, pilot FTZs can boost the optimization and upgrading of industrial structures by eliminating internal low-efficiency enterprises and absorbing advanced technologies from the outside [65]. With the continuous upgrading of the industrial structure, related enterprises also actively improve their production efficiency and achieve green production, which promotes the improvement of urban *GTFP*. Based on the above analysis, we propose hypothesis 2. In addition, the framework of influencing mechanisms of pilot FTZs on urban *GTFP* is shown in Figure 2.

**Hypothesis** **2** **(H2).***Pilot FTZs can increase GTFP through the promotion of industrial upgrading*.

## 3. Data and Methodology

### 3.1. The Measurement of GTFP

Urban *GTFP* can accurately describe and reflect the development efficiency of urban economies under the constraints of resources and the environment [39]. Under the framework of environmental technology analysis, this paper uses the slack-based measure (SBM) model based on undesirable output and the Malmquist index model to measure *GTFP*.

#### 3.1.1. SBM Model

The traditional data envelopment analysis (DEA) model has advantages in managing the problems of multi-input indices and multi-output indices [38,66]; thus, it is widely used in the field of development performance measurement. However, the traditional DEA method only considers economic benefits when measuring efficiency values but does not consider the impact of undesirable output. This fact is not only inconsistent with the actual production process but also ignores the slack problem of input and output factors; accordingly, there is a deviation between the measured value and the actual value [67]. Therefore, to solve this problem, Tone [68] first proposed a nonradial, nonangle SBM model based on slack variables. This model resolves the influence of slack variables on the measured value by incorporating the slacks of the input and output factors directly into the objective function, and it effectively improves the accuracy of the measured values [69,70]. Therefore, in line with the above research, the SBM model proposed in this paper is constructed as follows:(1)$$\rho =\underset{s^{x},s^{y},s^{b},\lambda }{\min}{\frac{1-\frac{1}{M}\sum_{m=1}^{M}\frac{s_{m}^{x}}{x_{km}}}{1+\frac{1}{N+L}\left(\sum_{n=1}^{N}\frac{s_{n}^{y}}{y_{kn}}+\sum_{l=1}^{L}\frac{s_{l}^{b}}{b_{kl}}\right)}}^{}$$
$$s.t.\left\{ \begin{array}{l} x_{km}=\sum_{k=1}^{K}\lambda_{k}x_{km}+s_{m}^{x},m=1,2,\ldots ,M\\ y_{kn}=\sum_{k=1}^{K}\lambda_{k}y_{kn}-s_{n}^{y},n=1,2,\ldots ,N\\ b_{kl}=\sum_{k=1}^{K}\lambda_{k}b_{kl}+s_{l}^{b},l=1,2,\ldots ,L\\ s_{m}^{x}\geq 0,s_{n}^{y}\geq 0,s_{l}^{b}\geq 0 \end{array}\right.$$
where ρ is the efficiency value; x, y, and b represent the vectors of the input factors, desirable outputs and undesirable outputs, respectively, and M, N and L represent the number of the types of input factors, desirable outputs and undesirable outputs, respectively; ($x_{km}$, $y_{kn}$, $b_{kl}$) are the input and output values of the *k*-th decision-making unit; *λ* represents the weight of the decision-making unit; and $s_{m}^{x}$, $s_{n}^{y}$, and $s_{l}^{b}$ are the slack variables of the input factors, desirable outputs and undesirable outputs, respectively. Among them, $s_{m}^{x}$ and $s_{l}^{b}$ represent the redundancy of input and undesirable output, respectively, and $s_{n}^{y}$ represents the shortage of desirable output; the objective function ρ is strictly monotonically decreases with respect to $s_{m}^{x}$, $s_{n}^{y}$, and $s_{l}^{b}$, and the value range of ρ is 0–1. If ρ=1, then $s_{m}^{x}=s_{n}^{y}=s_{l}^{b}=0$, which indicates that the evaluation unit is completely efficient, and there is no redundancy of input or shortage of output. If ρ=0, then the evaluation unit is technically completely inefficient at present.

In addition, Equation (1) represents the SBM model under the assumption of constant returns to scale (CRS). If the constraint condition $\sum_{k=1}^{K}\lambda_{k}=1$ is introduced into the SBM model, then Equation (1) can be transformed into the SBM model under the assumption of variable returns to scale (VRS).

The SBM model also has certain shortcomings. For example, its objective function is to minimize the efficiency value, that is, to maximize the inefficiency of input and output. From the perspective of the distance function, the projection point of the evaluated unit is the point on the front surface that is farthest from the evaluated unit. Tone [68] tried to design a method to calculate the shortest distance to the frontier, but it is only suitable for the case where there are fewer units to be evaluated. At present, the SBM model is still the mainstream method of efficiency measurement and has been widely used in the relevant literature [69,70,71,72].

#### 3.1.2. Malmquist Index Model

The Malmquist index is based on the distance function proposed by Malmquist in 1953 [73]. The index is calculated based on the quantitative data from inputs and outputs without the need for pricing information [74], and it can provide a breakdown of the change in productivity changes [75]. According to the Malmquist index model, we can accurately measure the dynamic growth trend of the *GTFP* in 280 cities of China. Referring to the practice of Chung et al. [76], the specific equation is set as follows:(2)$$GTFP=\left[\frac{\rho_{k}^{t}(x_{k}^{t},y_{k}^{t},b_{k}^{t})}{\rho_{k}^{t}(x_{k}^{t+1},y_{k}^{t+1})}\frac{\rho_{k}^{t+1}(x_{k}^{t},y_{k}^{t})}{\rho_{k}^{t+1}\left(x_{k}^{t+1},y_{k}^{t+1}\right)}\right]$$
where $\rho_{k}^{t}\left(x_{k}^{t},y_{k}^{t}\right)$ is the efficiency value of the decision-making unit in period t, $\rho_{k}^{t}\left(x_{k}^{t+1},y_{k}^{t+1}\right)$ is the efficiency value of the decision-making unit in period t+1, and $\rho_{k}^{t+1}\left(x_{k}^{t},y_{k}^{t}\right)$ $\rho_{k}^{t}\left(x_{k}^{t+1},y_{k}^{t+1}\right)$ are the cross-efficiency values in period t and period t+1, respectively. *GTFP* represents the green total factor productivity index. If *GTFP* > 1, then the *GTFP* of the decision-making unit increased in the current period, and if *GTFP* < 1, then the *GTFP* of the decision-making unit decreased in the current period.

However, the above indicator calculated according to the Malmquist index model is not the *GTFP* in each year but the rate of change of the *GTFP* relative to the previous year. Therefore, if we set the value of the *GTFP* in the base year to 1 and then multiply it by the *GTFP* of subsequent years, we can obtain the *GTFP* values of 280 sample cities in China from 2005 to 2017.

### 3.2. DID Method

In this paper, we use the DID method to evaluate the influence of FTZ policy on *GTFP*. The Chinese government established the China (Shanghai) Pilot FTZ in September 2013. To date, China has established pilot FTZs in six batches cover 21 provinces or municipalities. Therefore, we regard FTZ policy as a quasi-natural experiment. Tan and Yan [57] considered integrated FTZs to be the predecessor of pilot FTZs. To reduce the research bias, we regard the cities with integrated FTZs and pilot FTZs as the treatment groups. Other cities belong to the control group. The DID model is set as follows:(3)$${GTFP}_{it}=\alpha_{0}+\alpha_{1}{dt}_{it}+\sum \partial_{k}year_{k}+\sum \gamma_{j}X_{it}+\mu_{city}+\lambda_{it}$$
where *dt* represents whether city i has a pilot FTZ in year t and *GTFP* is the urban green total factor productivity measured by the super-efficiency SBM model of undesirable outputs and the Malmquist index. *X* represents a set of control variables, including the level of informatization of the city (Informatization), population density (Pdensity), the development level of the service industry (Service_sec), the level of science and technology (SciTec), and the scale of fixed assets (Fixed_assets). *year* represents the year fixed effect, *μ* represents the city fixed effect, and *λ* represents the random error term.

In addition, the premise of using a DID method is to meet the parallel trend hypothesis; that is, before the impact of the pilot FTZs, the *GTFP* of the experimental group and the *GTFP* of the control group have the same change trend. Referring to Li et al. [77] and Beck et al. [78], this paper uses the event analysis method and presents the dynamic model as follows:(4)$${GTFP}_{it}=\alpha +\sum_{k\geq -5,k\neq -1}^{5}\beta_{k}{D_{it}}^{k}+\sum \partial_{k}year_{k}+\sum \gamma_{j}X_{it}+\mu_{city}+\lambda_{it}$$
where *D_it_^k^* is a dummy variable that represents whether the city approved the establishment of a pilot FTZ. The value of *D_it_^k^* is assigned according to the following rules: *s_i_* represents the specific year of the establishment of a pilot FTZ. If *t* − *s_i_* < −5, then define *D_it_^k^* = 1; otherwise *D_it_^k^* = 0. If *t* − *s_i_* = *k*, then define *D_it_^k^* = 1; otherwise, *D_it_^k^* = 0 (*k*∈ [–5, 5] and *k* ≠ −1). The process of establishing a pilot FTZ is that the local governments submit an application to the State Council of China after completing the preparations and planning plans for pilot FTZs. The State Council then sends staff to the field to conduct research and perform repeated demonstrations and communication changes involving the specific plan. Only after more than a year of revisions and necessary administrative procedures will the State Council officially approve the establishment of a pilot FTZ. In fact, when the pilot FTZ was established, many service industry and high-tech companies had made preparations in advance. Therefore, we set the year before the pilot FTZ was approved as the base year.

### 3.3. Variables and Data Description

Dependent variable. As described in Section 3.1, the *GTFP* of 280 cities in China from 2005 to 2017 was estimated through the SBM model and the Malmquist index. The specific input factors, desirable outputs and undesirable outputs are presented in Table 2.

Dependent variable. As described in Section 3.1, *GTFP* is based on the DEA framework. This framework uses the non-radial and non-angle SBM model of undesirable outputs, and measures *GTFP* by constructing the Malmquist productivity index.

Independent variables. The FTZ policy (dt) is set in the form of a dummy variable. For the city with a pilot FTZ, the pilot year and subsequent years are set to 1, and the other years are set to 0. Up to now, China has established six batches of FTZs. the first batch of FTZs established in 2013 include the Shanghai FTZ, the second batch of FTZs established in 2015 include Guangzhou, Shenzhen, Zhuhai, Tianjin, Fuzhou, and Xiamen FTZs, and the third batch of FTZs established in 2017 includes Shenyang, Yingkou, Dalian, Zhoushan, Kaifeng, Luoyang, Zhengzhou, Yichang, Wuhan, Xiangyang, Chongqing, Chengdu, Luzhou and Xi’an FTZs. After that, the Chinese government successively established the fourth, fifth and sixth batches of free trade zones in different cities. Due to data limitations, this paper regards only the cities with pilot FTZs in the first, second, and third batches as the research objects.

Control variables. This paper includes the following control variables: the level of informatization of the city (Informatization), as measured by the number of internet users in the city (by taking the logarithm); population density (Pdensity), measured by the number of people per square kilometer; the development level of the service industry (Service_sec), as measured by the proportion of the added value of the service industry in GDP; the level of science and technology (SciTec), as measured by the amount of urban technology investment (by taking the logarithm); and the scale of fixed assets (Fixed_assets), as measured by the amount of fixed asset investment in the city (by taking the logarithm).

The data source is the “China Urban Statistical Yearbook”. The sample of this paper consists of 2005–2017 data on 280 prefecture-level cities in China. The descriptive statistics of each variable are presented in Table 3.

## 4. Results and Discussion

### 4.1. Parallel Trend Test and Dynamic Test

Before using the DID method to evaluate the impact of FTZ policy on *GTFP*, we first conducted a parallel trend test based on Equation (4). The changing trend of *GTFP* before and after the implementation of FTZ policy is shown in Figure 3. The figure illustrates that before the construction of pilot FTZs, there was no significant difference between the experimental and control groups. Therefore, the parallel trend hypothesis is met, and it is reasonable to use the DID method to evaluate the impact of FTZ policy on *GTFP*.

Figure 3 also allows us to determine the dynamic effect of FTZ policy on *GTFP*. The results illustrate that the driving effect of pilot FTZs on *GTFP* is significant in the first and second years. This indicates that pilot FTZs effectively attracted the agglomeration of environmentally friendly foreign capital in the zones, and supportive policies enabled the development of some high-tech industries and green industries, which, in turn, promoted urban green development. However, we also find that in the third, fourth, and fifth years after the establishment of pilot FTZs, the promotion effect was no longer significant. This may be because imperfect policy guarantee systems, especially the failure to establish an effective intellectual property system, interfered with the positive impact of pilot FTZs on *GTFP* [49]. We can see that the effect of pilot FTZs on *GTFP* shows an upward trend from the third year. Due to the unavailability of data, we are not able to estimate the long-term effect of pilot FTZs on *GTFP*. Since the ideas of green development are continuously applied in the pilot FTZs, we can infer that a long-term positive effect will exist. In the future, the governments should continue to adhere to the ideas of green development in the pilot FTZs so that they can effectively drive urban green development.

### 4.2. Baseline Regression

Table 4 shows the baseline regression results of the influence of FTZ policy on *GTFP*. In column (1), only the dummy variable (dt) is used as the independent variable for the regression analysis, and the estimated coefficient is positive and significant at the 1% level. There are three possible reasons for this result. First, there is the time trend effect. Second, there is selectivity bias; that is, the cities where pilot FTZs are located have a higher *GTFP*. Third, the FTZ policy significantly improved the *GTFP* of cities. Column (2) adds time dummies to the control variables. The regression coefficient remains significantly positive, and the regression results show that the urban *GTFP* level has a trend of increasing year by year excluding the first reason above. Column (3) further controls for city-specific fixed effects, and the regression coefficient is significantly positive at the 1% level. This result shows that after controlling for the differences between the experimental and control groups, the regression coefficient of *dt* is still significantly positive at the 1% level, which further excludes the second reason and indicates that the pilot FTZ policy has significantly improved *GTFP*. The control variables are added in column (4), and the results do not change substantially.

### 4.3. Regression Analysis Based on PSM-DID

To minimize the systematic differences between the experimental and control groups as much as possible and to reduce the estimation bias of the DID model, referring to Liu and Zhao [82], this paper further uses the PSM-DID approach to test the robustness of the above results. The propensity score is obtained by conducting a logit regression on a series of variables for cities with or without a pilot FTZ. The city with the propensity scores closest to that of the experimental group (the city with a pilot FTZ) is regarded as the control group. After obtaining the matched experimental group and control group cities, it is still necessary to further verify whether the matching results meet the common support assumption. The results after matching are presented in Figure 4 and Table 5, which show the reduced bias in the covariates after matching, and that most samples are successfully matched.

This section describes use of the PSM-DID method as a robustness test to study the effects of the FTZ policy on *GTFP*. The estimated results are presented in Table 6, which illustrates that the impact of the FTZ policy on *GTFP* is still significantly positive after using the PSM-DID method. The differences in the regression coefficients are not substantial, and the significance is almost the same whether using the DID method or using the PSM-DID method, which verifies that the policy is beneficial in improving *GTFP*.

The present research contributes to the literature on policy zones that are devoted to promoting FDI, international trade, and the environment. Moreover, on the basis of previous research concerning the effect of pilot FTZs on economic growth [31,32] and environmental pollution [36], this paper further shows that the urban economic growth brought by the pilot FTZs is not at the expense of the environment. Using DID and PSM-DID, our findings are consistent with the research conclusion of Jiang et al. [18] based on the case of the Shanghai pilot FTZ. This suggests that the concept of green development applied by the central and local governments in pilot FTZs is effective, which is not just limited to Shanghai, and the FTZs as green zones drive the green development of cities.

### 4.4. Heterogeneity Analysis

We analyzed the heterogeneity from the aspects of location and city size. There are numerous differences in the level of economic development, marketization, and institutional quality among the various regions in China [83]. As a result, we can expect that pilot FTZ policy may have a different promoting effect on *GTFP* in different regions. Following the research of Gong and Shen [84], we divided the samples into three categories, namely, the eastern, central, and western regions, to explore the heterogeneous effects of the FTZ policy on *GTFP*. The results are shown in Columns 1–3 of Table 7. The construction of pilot FTZs significantly promote the level of *GTFP* in central and western China, but the impact on cities in eastern China is not significant. The levels of trade openness, industrial development, and technological innovation in the eastern region are inherently high; therefore, the marginal effect of the FTZ policy on *GTFP* is not significant. Although the development level is lower in the central region than in the eastern region, the central region has relied on the support of national policies to seek opportunities for development in recent years. The pilot FTZ policy can effectively promote trade facilitation and the entry of high-quality foreign capital in the central region, and thus greatly promote the upgrading of the industrial structure and the green innovation of enterprises. As a region with relatively backward economic development in China, the optimization of trade administration and the facilitation of trade and investment in the pilot FTZs in the western region have released a large institutional dividend, which has a significant marginal effect on *GTFP*.

In general, compared with smaller cities, larger cities tend to have advantages in industrial structure, resource agglomeration, and scientific and technological development [85,86], which may render the policy effect of pilot FTZs on *GTFP* heterogeneous. Therefore, we also carried out a subsample regression according to the different sizes of cities. According to the size of the urban population, we divided the sample into large and medium-sized cities and small cities. If the population of a city is more than 500,000, we defined it as a large and medium-sized city, and if the population of a city is less than 500,000, then we defined it as a small city. The regression results are shown in Columns 4 and 5 of Table 7. In large and medium-sized cities, pilot FTZs have significantly promoted the improvement of *GTFP*, while in small cities, the policy effect has not been significant. This difference may be because large and medium-sized cities have not only good basic conditions for trade and foreign investment but also a high degree of spatial agglomeration of innovation [85]. Therefore, the pilot FTZ policy can significantly promote the development of green technology. However, in small cities, the level of capital stock is weak, the industrial foundation is poor, and the level of scientific and technological development is low. As a result, pilot FTZs fail to have good policy effects in the short term in small cities.

## 5. Exploring the Mediating Effect of the FTZ Policy on *GTFP*

To explain the potential mechanisms of the green economic growth impacts, we believe that pilot FTZs can enhance *GTFP* by promoting FDI and industrial upgrading. To test this hypothesis, according to Baron and Kenny [87], we constructed a mediating effect model:(5)$$GTFP_{it}=\beta_{0}+\beta_{1}{dt}_{it}+\sum \delta_{k}year_{k}+\phi \sum_{N=1}^{N}X_{it}+\mu_{city}+\nu_{it}$$
(6)$$M_{it}=\lambda_{0}+\lambda_{1}{dt}_{it}+\sum \delta_{k}year_{k}+\theta \sum_{N=1}^{N}X_{it}+\mu_{city}+\xi_{it}$$
(7)$$GTFP_{it}=\gamma_{0}+\gamma_{1}{dt}_{it}+\gamma_{2}M_{it}+\sum \delta_{k}year_{k}+\psi \sum_{N=1}^{N}X_{it}+\mu_{city}+\tau_{it}$$

In Equations (5)–(7), *M* represents the mediating variable, which is defined from the perspective of FDI and industrial upgrading. FDI is measured by the amount of FDI in a city. Referring to Gan et al. [88], industrial structure upgrading (INSU) is measured by the ratio of the output value of the tertiary industry to the output value of the secondary industry.

The steps in testing the mediating effect are as follows. First, urban *GTFP* is considered to be the dependent variable, and the pilot FTZs are taken as the core independent variable for the regression. Second, the mediating variables are considered to be the dependent variables, and the pilot FTZs are taken as the independent variable for the regression. Finally, the pilot FTZs variable and mediating variables are both included in the regression model to observe their impacts on urban *GTFP*. If the coefficients $\beta_{1},\;\lambda_{1},\;\gamma_{2}$ are significant and $\gamma_{1}$ decreases or is significantly lower than $\beta_{1}$, then a mediating effect exists.

Column 1 in Table 8 presents the overall effect of the FTZ policy on *GTFP* based on Equation (5), which suggests that pilot FTZs significantly improve *GTFP*. This result might be because the pilot FTZs expand the scale of FDI and promote industrial structure upgrading, thereby enhancing *GTFP*. As shown in Column 2 in Table 8, we find a positive and statistically significant impact of the FTZ policy on FDI. Turning to the results in Column 3, the estimated coefficients of FDI and the pilot FTZs are both statistically significant. Compared with the results in Column 1, the regression coefficient of the pilot FTZs decreases, which implies that this policy has promoted an increase in *GTFP* by expanding the scale of FDI. By the same logic, the regression results in Columns 1, 4 and 5 indicate that pilot FTZs can enhance *GTFP* by promoting industrial structure upgrading.

Although several studies, such as Jiang et al. [18], have examined the impact of the Shanghai pilot FTZ on green development, they did not study the underlying mechanism. The present study fills this gap and finds that pilot FTZs promote the green development of cities by promoting FDI and industrial structure upgrading. First, government has implemented a series of institutional innovations that improve investment liberalization and facilitation in the pilot FTZs, which has greatly encouraged the entry of foreign capital. It is worth noting that pilot FTZs have strict environmental requirements for foreign investment; therefore, they mainly attract foreign investment with high-tech, high value-added, and environmentally friendly characteristics, which is conducive to the green development of a city. In addition, the government attaches great importance to the industrial planning and design of pilot FTZs and is committed to prioritizing the development of the high-tech industry, high-end service industry, and financial services industry to promote industrial upgrading and green development.

## 6. Conclusions

To promote an all-dimensional, multilayered, and wide-ranging opening up to the outside world, China set up pilot FTZs after the establishment of SEZs, EPZs, and BZs. The Chinese pilot FTZs not only facilitate trade and investment liberalization and facilitation through institutional innovation but also take green development as the basic concept in the development of construction plans. For this reason, this research investigates the impact of FTZ policy on the green and healthy development of cities by using the DID and PSM-DID methods. Based on a theoretical analysis, we find that pilot FTZs have strict environmental requirements for foreign investment. In particular, pilot FTZs mainly encourage foreign investment with high-tech, high value-added, and environmentally friendly characteristics, which is beneficial to the green growth of a city. In addition, the industry planning and design sets for pilot FTZs tend to prioritize the development of the high-tech industry, high-end service industry, and financial services industry, which promotes industrial upgrading and further boosts the green development of cities. Consequently, we propose the hypothesis that pilot FTZs promote green development through FDI and industrial upgrading and confirm the mechanism through the mediation model.

The evidence indicates that, first, the pilot FTZs have significantly improved the *GTFP* of cities, and the results are robust to the estimation conclusion of the PSM-DID. Second, pilot FTZs have a positive impact on urban *GTFP* by expanding the scale of FDI and promoting industrial structure upgrading. Third, there are heterogeneous effects according to the region and city size. The impact of FTZ policy on *GTFP* is significant in the central and eastern cities of China but not in the western regions. In addition, pilot FTZs have a positive effect in large- and medium-sized cities, while their effects are not significant in small cities.

The research conclusions of this article suggest that pilot FTZs carry out trade and investment following internationally accepted green trade rules and commercial and ecological environmental management rules. The concept of green development penetrates the entire process of the pilot FTZs construction. In addition, the pilot FTZs vigorously develop the modern green service industry, green manufacturing, and green supply chain, which greatly promotes the green and healthy development of cities. Indeed, the pilot FTZs aim to develop into green areas, which has become a feature that distinguishes them from Chinese policy zones such as SEZs, EPZs, and BZs and the FTZs in other countries [11].

European countries are at the forefront in the fields of green energy, low-carbon technology, and environmental governance technologies [89]. China urgently needs the EU’s advanced technical support and experience sharing to takes advantage of the considerable potential in the clean energy market and low manufacturing costs. At present, some pilot FTZs such as the Qingdao pilot FTZ have cooperated with Germany and other countries on sustainable production. In the future, the Chinese pilot FTZs can further relax market access for EU companies in the above fields and attract them to enter China’s environmental protection market. More importantly, pilot FTZs should actively use cooperation with European countries in these fields to boost the green development of cities.

Academic research in the future should give attention to how the pilot FTZs can adhere to the concept of green development and continue to transform themselves into green areas in the future. Due to data limitations, our research can only examine the short-term effect of FTZs on urban green development. With more data available, future researchers will have the opportunity to explore the long-term effect of the pilot FTZs on urban green development. Moreover, evaluating the development status of green finance, environmentally conscious manufacturing, and environmental supply chains, etc., in FTZs is also an important research topic in the future.

## Figures and Tables

**Figure 1 ijerph-18-11681-f001:**
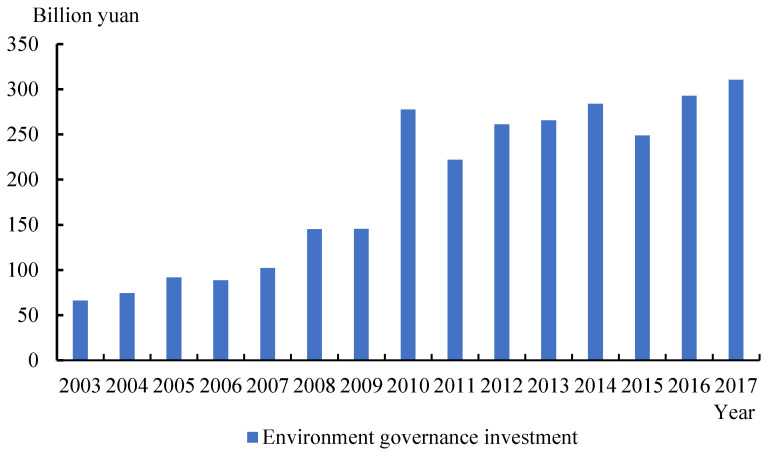
The average value of environmental governance investment in provinces where pilot FTZs are located. Source: China Statistical Yearbook on Environment, 2003–2017.

**Figure 2 ijerph-18-11681-f002:**
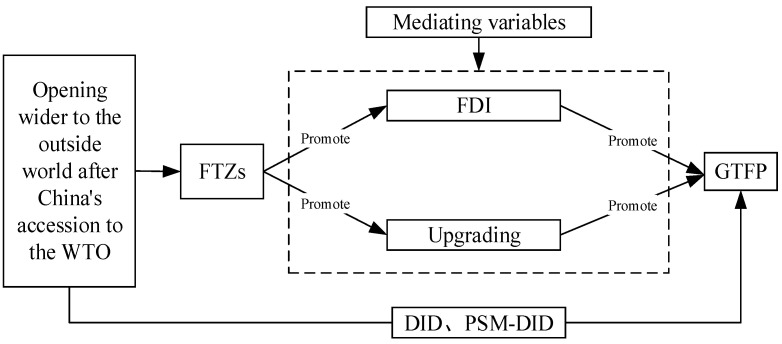
The framework of the influence of pilot FTZs on *GTFP*. Source: Compiled by the author.

**Figure 3 ijerph-18-11681-f003:**
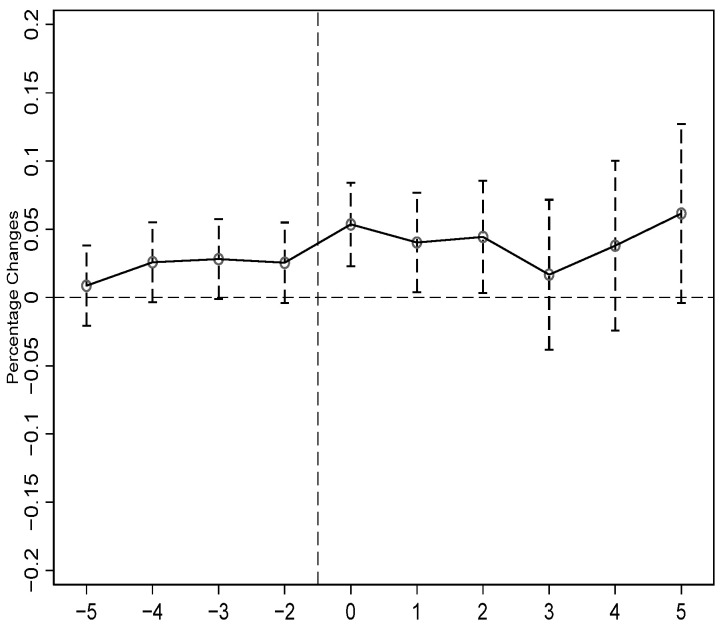
Parallel trend and dynamic effect test of the impact of FTZ policy on *GTFP*.

**Figure 4 ijerph-18-11681-f004:**
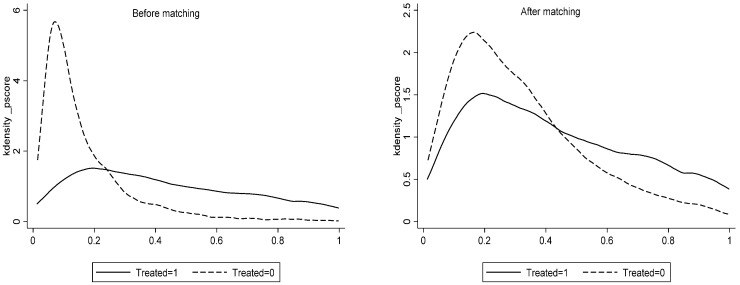
Kernel distribution of the propensity scores before and after matching.

**Table 1 ijerph-18-11681-t001:** Comparison between FTZs and other policy zones.

Policy	Industry	Free Currency	Free Trade	Free Entry and Exit of Goods	Free Storage of Goods	Free Entry and Exit of Personnel
FTZs	√	√	√	√	√	√
SEZs	√	×	×	×	×	×
EPZs	√	×	×	×	×	×
BZs	√	×	×	√	√	×

Source: The author manually compiled according to the relevant policies of China. √ means that the innovative policies such as industry policy, free currency policy, and free entry and exit of goods, etc. were implemented in the policy zones; In contrast, × represents the policies were not implemented in the policy zones.

**Table 2 ijerph-18-11681-t002:** Explanation of each indicator.

Category	Factor	Explanation
Input factors	Capital stock	Based on the calculation method provided by Shan [79], the perpetual inventory method is used to measure capital stock. The depreciation rate is set to 10.96%, and the fixed asset investment data for each city are deflated by using the fixed asset investment price index of the province. As a result, we adjust it to the actual value of constant prices in 2005.
	Number of laborers	The total number of employees in the secondary and tertiary industries is used as an indicator of the labor input in various cities.
	Energy input	The entire society’s electricity consumption and urban water supply are used as measures of energy and resource consumption, respectively, during economic development.
Desirable outputs **	Urban output	Actual regional GDP
	Resident quality of life	Green coverage of the built-up area
Undesirable outputs	Wastewater	Industrial wastewater discharge
	Exhaust gas	Industrial SO_2_ emissions ***
	Soot	Industrial soot emissions
	Haze pollution	PM2.5 concentration *

Data source: The above data are mainly from the “China Urban Statistical Yearbook” and the statistical bulletins of provinces and cities over the sample period. * The PM2.5 concentration data for cities come from satellite remote sensing data compiled by the National Aeronautics and Space Administration (NASA) (http://earthdata.nasa.gov (accessed on 1 November 2020). We use the 1.4 million Chinese basic geographic information data provided by the National Basic Geographic Information Center to obtain the average PM2.5 concentration in various cities over the sample period [80]. ** Desirable outputs refer to strong (good) outputs, such as paper or electricity. Undesirable outputs refer to weak (bad) outputs, such as suspended solids or SO_2_ [76,81]. *** Since the Chinese official government has not published the city-level CO_2_ industrial emissions data, the industrial SO_2_ emissions is adopted here.

**Table 3 ijerph-18-11681-t003:** Descriptive statistics.

	Variables	N	Mean	Std	Min	Max
Dependent variable	*GTFP*	3640	0.369	0.222	0.0138	3.679
Independent variables	dt	3640	0.0555	0.229	0	1
	treated	3640	0.236	0.425	0	1
Mediating variables	FDI	3481	0.761	1.853	0.0003	30.83
	INSU	3591	0.846	0.426	0.0943	4.166
Control variables	Informatization	3606	12.69	1.160	5.466	17.76
	Pdensity	3566	0.436	0.325	0.0047	2.648
	Service_sec	3591	37.56	9.048	8.580	80.23
	SciTec	3623	9.469	1.709	3.526	15.21
	Fixed_assets	3596	15.62	1.148	11.83	18.97

**Table 4 ijerph-18-11681-t004:** Baseline regression results.

Variables	(1)	(2)	(3)	(4)
	*GTFP*	*GTFP*	*GTFP*	*GTFP*
dt	0.098 ***	0.053 ***	0.058 ***	0.034 ***
	(0.012)	(0.012)	(0.013)	(0.013)
Informatization				−0.021 ***
				(0.007)
Pdensity				0.319 ***
				(0.071)
Service_sec				0.002 ***
				(0.001)
SciTec				0.016 ***
				(0.005)
Fixed_assets				−0.045 ***
				(0.009)
Year FE	NO	YES	YES	YES
City FE	NO	NO	YES	YES
Constant	0.363 ***	0.390 ***	0.390 ***	0.954 ***
	(0.002)	(0.013)	(0.008)	(0.151)
Observations	3640	3640	3640	3529
R^2^	0.0202	0.0568	0.0568	0.0755

***, **, and * indicate statistical significance at the 1%, 5% and 10% levels, respectively. * To save space, this paper does not present the regression results of the year dummy variables in Table 4.

**Table 5 ijerph-18-11681-t005:** Reduced bias in the covariates after matching.

Variables	State	Mean (Treated)	Mean (Control)	% Bias	T	P
Informatization	Unmatched	13.581	12.404	112.3	28.44	0.000
	Matched	13.535	13.527	0.7	0.15	0.880
Pdensity	Unmatched	0.62111	0.37941	75.8	19.75	0.000
	Matched	0.60123	0.67936	−24.5	−3.31	0.001
Service_sec	Unmatched	43.24	35.641	85.0	23.04	0.000
	Matched	42.675	42.239	4.9	0.99	0.320
SciTec	Unmatched	10.711	9.0694	100.9	26.70	0.000
	Matched	10.625	10.56	4.0	0.86	0.389
Fixed_assets	Unmatched	16.565	15.317	125.3	30.94	0.000
	Matched	16.534	16.474	6.0	1.36	0.173

**Table 6 ijerph-18-11681-t006:** PSM-DID estimation results.

Variables	(1)	(2)	(3)	(4)
	*GTFP*	*GTFP*	*GTFP*	*GTFP*
dt	0.092 ***	0.039 ***	0.045 ***	0.022 *
	(0.012)	(0.013)	(0.013)	(0.013)
Informatization				−0.023 ***
				(0.007)
Pdensity				0.254 ***
				(0.076)
Service_sec				0.002 ***
				(0.001)
SciTec				0.007
				(0.006)
Fixed_assets				−0.048 ***
				(0.011)
Year FE	NO	YES	YES	YES
City FE	NO	NO	YES	YES
Constant	0.355 ***	0.364 ***	0.359 ***	1.088 ***
	(0.002)	(0.014)	(0.009)	(0.177)
Observations	3295	3295	3295	3184
R^2^	0.0197	0.0595	0.0595	0.0734

***, **, and * indicate statistical significance at the 1, 5 and 10% levels, respectively.

**Table 7 ijerph-18-11681-t007:** Heterogeneity analysis of the effects of FTZ policy on *GTFP*.

Variables	(1)	(2)	(3)	(4)	(5)
	Location			City Size	
	Eastern	Central	Western	Large and Medium-Sized Cities	Small Cities
dt	0.013	0.036 *	0.085 **	0.032 **	−0.018
	(0.019)	(0.019)	(0.043)	(0.013)	(0.092)
Constant	0.789 ***	0.432 **	1.657 ***	0.993 ***	0.933 ***
	(0.275)	(0.201)	(0.390)	(0.204)	(0.243)
Controls	YES	YES	YES	YES	YES
Year FE	YES	YES	YES	YES	YES
City FE	YES	YES	YES	YES	YES
Observations	1439	1346	744	2415	1114
R^2^	0.116	0.069	0.095	0.097	0.071

***, **, and * indicate statistical significance at the 1, 5 and 10% levels, respectively. * The classification of city size comes from the Notice of the State Council on Adjusting the Standards for Categorizing City Sizes, No. 51 [2014], State Council of China.

**Table 8 ijerph-18-11681-t008:** Transmission mechanism of the FTZ policy affecting *GTFP*.

Variables	(1)	(2)	(3)	(4)	(5)
	*GTFP*	FDI	*GTFP*	INSU	*GTFP*
dt	0.034 ***	0.631 ***	0.021 *	0.049 ***	0.032 **
	(0.013)	(0.072)	(0.013)	(0.009)	(0.013)
FDI			0.018 ***		
			(0.003)		
INSU					0.048 ***
					(0.016)
Constant	0.954 ***	−1.251	0.918 ***	0.813 ***	0.965 ***
	(0.151)	(0.884)	(0.154)	(0.105)	(0.150)
Controls	YES	YES	YES	YES	YES
Year FE	YES	YES	YES	YES	YES
City FE	YES	YES	YES	YES	YES
Observations	3529	3392	3392	3529	3529
R^2^	0.075	0.187	0.084	0.788	0.075

***, **, and * indicate statistical significance at the 1, 5 and 10% levels, respectively.

## Data Availability

The data presented in this study are available on request from the corresponding author. The data are not publicly available due to privacy.

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
