# Peer review of "Have China’s Pilot Free Trade Zones Improved Green Total Factor Productivity?"

_ijerph, 2021, doi:10.3390/ijerph182111681_

Round 1

Reviewer 1 Report

Page 1, Introduction, paragraph 2: you must provide a complete definition of FTZ. It’s not just about ‘designated areas where goods can be stored, shipped in or out, manufactured or handled under a set of specific customs regulations’. There are specific tax regulations (fiscal incentives in the form of tax holidays and reduced tax regime), plus a one-window service, etc. Generally, more clarification is needed there.

Page 3: “On September 27, 2013, the first Pilot Free Trade Zone” – this is not very true. Are authors talking about specific ‘green’ FTZ or in general? If in general, authors must know that FTZ, EPZs, SEZs have been actively implemented in China since the 70s. So, the authors need to rephrase this statement, delete it, or clarify what they mean better. Or they must provide a summary of special economic zones development in China since the 70s.

Page 4. Little has been said that SEZs and FTZ must be ideally based on cluster policy to gain other benefits as participation in global value chains, knowledge exchange, attracting more investors around the anchor one, innovation stimulation, etc. Also, the role of the government in the development of the FTZs. This is particularly relevant for the Chinese case. The authors need to include this discussion in their literature review.

Page 2. Figure 1. It looks clumsy What do those brackets [] mean? Also, I don’t think that this figure is necessary. It can be excluded.

Page 4, Figure 2. is very clumsy. It needs to look more professional. It’s not clear: is it millions or thousands? What currency? Revise all figures in your work.

Page 6, Figure 1 (again?). Why is the third figure actually ‘Figure 1’? Also, what is the source of it?

Page 6. It’s not clear where the proposed model is coming from. The authors need to justify their logic of selecting or constructing this model with relevant citations.

Page 8: “In this paper, we use the DID mode”. I don’t understand what the point was to show the previous two models. I’d recommend the authors delete the discussion of the first two and focus more on the justification of the DID model if this was the one that the authors had used. Or, if the first two models explain the logic of selecting the third one, this should be better clarified.

Page 7, Table 1 should be placed in ‘3.3. Variables and data description’.

Page 9. “4.1. Parallel trend test and dynamic test” – further clarification of the used formula should be placed in Methodology. Authors should focus on Empirical results only.

Page 10. Figure 2 should be Figure 3. The authors need to look at the numbering of the figures throughout the paper again.

This paper doesn’t have a Discussion section. It either should be placed within the Findings chapter, or a new section should be created. In this section, authors need to clarify how their findings contrast or supplement the existing studies – those that they provided in their Literature Review. Hence, this Discussion should contain relevant citations.

The conclusion is quite generic. The authors should start with the initial aim of the paper and then go through their hypotheses and explain what they have found in their study according to them. Recommendations for future research can be developed further: the phrase “future studies can be further conducted from the perspective of other channels” is unclear. 

Author Response

Dear reviewer,

Thank you for your letter and for the editor and reviewers’ comments concerning our manuscript entitled “Have China’s pilot free trade zones improved green total factor productivity? Fresh evidence from a quasi-experimental approach” (ijerph-1392324). Those comments are all valuable and very helpful for revising and improving our paper. The responses to the comments are given below.

Reviewer 1:

Comments to the Author

Comment 1:

Page 1, Introduction, paragraph 2: you must provide a complete definition of FTZ. It’s not just about ‘designated areas where goods can be stored, shipped in or out, manufactured or handled under a set of specific customs regulations’. There are specific tax regulations (fiscal incentives in the form of tax holidays and reduced tax regime), plus a one-window service, etc. Generally, more clarification is needed there.

Response 1:

Thank you very much for the positive comments and constructive suggestions. We re-defined the concept of the FTZ, the specific content is as follows:

According to Teifenbrun (2015) and Akbari et al. (2019), FTZ refers to an area where goods may be landed, handled, manufactured, or reconfigured, and then re-exported without the intervention of customs authorities, and multiple institutional incentives such as free movement of goods and people, and preferential taxation are provided in it. Since the 1980s, countries such as the United States and the Brazil have established the FTZs, and they are conductive to attract Foreign Direct Investment (FDI), and greatly promote trade development (Chauffour et al., 2011; Benton, 2016; Mohebi and Mirshojaee, 2019). And researchers have found that FTZs play an important role in enhancing the competitiveness of enterprises in zones and promoting economic growth (Kouparitsas, 1997; Feils, 2008; Seyoum, 2012; Castilho et al., 2018). The Chinese government has always been committed to deepening reform and opening wider to the outside world. After establishing special economic zones (SEZs), export processing zones (EPZs), and bonded zones (BZs), Chinese government further implemented the FTZ policy. Compared to the previous policy zones like SEZs, the opening-up of the service and financial industries has been expanded in the FTZs for the first time. And the Chinese FTZs have also implemented a series of institutional innovations except for in possession of the functions similar to those of other countries. The institutional innovations mainly include, first, optimizing government functions, and improving trade and investment efficiency through one-window service, etc. Second, issuing tax holiday and tax regime such as reduced income tax for some enterprises in FTZs. Third, introducing the negative list management model, and foreign capital is allowed to enter in areas which are not in the list.

Reference

Akbari M, Azbari M E, Chaijani M H. Performance of the Firms in a Free‐Trade Zone: The Role of Institutional Factors and Resources[J]. European Management Review, 2019, 16,363-378.

Benton C N, Napier M, Uelkue M A. On Supply Chain Integration to Free Trade Zones: The Case of the United States of America[J]. Global Business Review, 2016, 17(4):1-11.

Castilho, M., Menéndez, M. & Sztulman, A. Poverty changes in Manaus: Legacy of a Brazilian FTZ? [J]. Review of Development Economics,2018,23(1) :102-130.

Chauffour, J. P., Maur, J. C. Preferential trade agreement policies for development: a handbook[J]. World Bank Publications, 2011.

Feils, D.J., Rahman, M. Regional economic integration, and foreign direct investment: The case of NAFTA[J]. Management International Review, 2008, 48 (2): 147 -163.

Mohebi M, Mirshojaee F. Microdata analysis of the consequences of free trade port policy: the case of Iran free zones[J]. Applied Economics, 2019(1):1-11.

Kouparitsas, MA. A dynamic macroeconomic analysis of NAFTA[J]. Economic Perspectives.1997, 21, 14-35.

Seyoum, B., Ramirez, J. Foreign trade zones in the United States[J]. Journal of Economic Studies, 2012, 39,13-30.

Teifenbrun, Susan (2015) "U.S. Foreign Trade Zones and Chinese Free Trade Zones: A Comparative Analysis," [J]. Journal of International Business and Law,14,2.

Comment 2:

Page 3: “On September 27, 2013, the first Pilot Free Trade Zone” – this is not very true. Are authors talking about specific ‘green’ FTZ or in general? If in general, authors must know that FTZ, EPZs, SEZs have been actively implemented in China since the 70s. So, the authors need to rephrase this statement, delete it, or clarify what they mean better. Or they must provide a summary of special economic zones development in China since the 70s.

Response 2:

Thank you very much for your recommendation. We have sorted out the development process from SEZs, EPZs, and BZs to FTZs, and the specific revisions are as follows:

Since 1978, China embarked on the great course of reform and opening up. In 1980s, Chinese central government first began to establish SEZs and launched a series of preferential policies for FDI introduction. And then, to further attract FDI and develop export-oriented industries, EPZs were successfully constructed in some coastal areas. As the 21st century got underway, China has successively established a series of BZs, in the zones, goods can freely enter and exist. The above-mentioned policy zones have played an important role in the FDI introduction and free trade development in China. However, there is still so much room for China to expand the opening of its service industry and attract advanced foreign investment.

Comment 3:

Page 4. Little has been said that SEZs and FTZ must be ideally based on cluster policy to gain other benefits as participation in global value chains, knowledge exchange, attracting more investors around the anchor one, innovation stimulation, etc. Also, the role of the government in the development of the FTZs. This is particularly relevant for the Chinese case. The authors need to include this discussion in their literature review.

Response 3:

It is really true as Reviewer suggested that we should include more discussion on SEZs and FTZs in literature review, and we need to emphasize the role of the government in the development of the FTZs. Therefore, we added the following content:

Numerous studies have examined the economic effects of the policy zones. Some literature found that Chinese SEZs, EPZs, and BZs are all conducive to trade development, attracting more investors and FDI (Hong et al., 2016; Yang et al., 2017; Zhang et al.,2019). And these special zones gain other benefits as participation in global value chains, knowledge exchange, and innovation stimulation, etc. (Miyagiwa, 1986; Sun and Zhang, 2019). Some researchers indicated that the FTZs also have effects in attracting FDI and promoting trade development (Li et al., 2021). And Jiang et al. (2021) believed that FTZs can effectively combine domestic production factors with international advanced technology, thus driving innovation. And FTZs greatly promote cross-country knowledge spillovers, improve China’s competitive advantages in global industrial and supply chains (Sheng, 2015; Yang and Xiao, 2021). Additionally, Song et al. (2018) found the policy advantages of FTZ is conductive to upgrading the organization and performance of local enterprises.

Reference

Hong, C, Nan, L, He, Y. Remanufacturing of electronic products in bonded port area across home and foreign markets[J]. The International Journal of Logistics Management, 2016, 27(2):309-334.

Jiang Y, Wang H, Liu Z. The impact of the free trade zone on green total factor productivity ——evidence from the shanghai pilot free trade zone. Energy Policy [J]., 2021, (148).

Li S, Liu J, Kong Y. Pilot Free Trade Zones and Chinese Port-listed companies Performance: An Empirical Research Based on Quasi-Natural Experiment[J]. Transport Policy, 2021.

Miyagiwa K F. A reconsideration of the welfare economics of a free-trade zone[J]. Journal of International Economics, 1986, 21(3-4):337-350.

Sheng, B. Tianjin Free Trade Zone: a comprehensive experimental field for institutional innovation[J],2015(1):4-10.

Yang, J, Zhou, L, Zou H. Evaluation of the Economic Growth Effect of the Establishment of Special Economic Zones in my country——Analysis Based on the Synthetic Control Method[J]. Economic Perspectives,2017(01):41-51.

Yang, Z, Xiao, J. Optimized path of regional industrial supply chain development under the background of free trade zone construction[J]. Journal of Commercial Economics,2021(9):176-178.

Song, M, Wang, J, Wang S, et al. Knowledge accumulation, development potential and efficiency evaluation: an example using the Hainan free trade zone[J]. Journal of Knowledge Management, 2018.

Sun, P, Zhang, Y. Foreign Investment Opening Policy, Export Processing Zone and Export Survival——From the Perspective of Industrial Linkage Effect[J]. China Economic Quarterly,2019,18(2):701-720

Zhang, P, Xu, J, Liu, H. The Effectiveness of Industrial Policy in Promoting Global Value Chain Upgrading: Based on the Quasi-Experiment of Chinese Export Processing Zones[J]. Journal of Financial Research, 2019(05):76-95.

Comment 4:

Page 2. Figure 1. It looks clumsy What do those brackets [] mean? Also, I don’t think that this figure is necessary. It can be excluded.

Response 4:

We are grateful for the suggestion. To be more accuracy and in accordance with the reviewer concerns, we have deleted Figure 1 in the original manuscript. And we add a brief description in manuscript as follows: “According to Report on the State of the Ecology and Environment, in China 2020, 40.1% of the 337 cities at or above prefecture-level exceeded air quality standards”.

Comment 5:

Page 4, Figure 2. is very clumsy. It needs to look more professional. It’s not clear: is it millions or thousands? What currency? Revise all figures in your work.

Response 5:

Thank the reviewer for the constructive comments and suggestions. We set up the horizontal axis with units of year, which represents time, and the vertical axis with units of billion yuan (RMB), which represents environmental governance investment to make Figure 2 more professional. And it’s worth noting that we have changed the order of figures, Figure 2 in the original manuscript is now Figure 1.

Comment 6:

Page 6, Figure 1 (again?). Why is the third figure actually ‘Figure 1’? Also, what is the source of it?

Response 6:

Thanks for your kind suggestion, we’re sorry that our carelessness has caused you inconvenience in reading the previous manuscript. We have revised and updated the order of figures in the latest version of the paper. In addition, regarding the source of this Figure, it is compiled by the author according to two mechanism routes “FDI” and “Upgrading”. The revised figure is shown as follows:

Figure 1. The framework of the influence of pilot FTZs on GTFP. Source: Compiled by the author.

Comment 7:

Page 6. It’s not clear where the proposed model is coming from. The authors need to justify their logic of selecting or constructing this model with relevant citations.

Response 7:

We appreciate for your valuable comment. According to your suggestion, we have added the source of SBM model. The specific content is as follows:

Tone (2001) first proposed a non-radial, non-angle SBM model based on slack variables. The model resolves the influence of slack variables on the measured value by incorporating the slack variables of input and output factors directly into the objective function, and it effectively improves the accuracy of the measured values (Pongpanich et al., 2017; Huang et al., 2020). And in line with the above research, the SBM model, proposed in this paper, is constructed as follows.

Reference

Huang Q, Hu J, Chen X. Environmental regulation, and green total factor productivity: dilemma or win-win? [J]. China Population, Resources and Environment, 2018, 28(11):140-149.

Pongpanich, R, Peng, et al. The performance measurement of listed companies of the agribusiness sector on the stock ex-change of Thailand[J]. Agricultural Economics, 2017, 63.

Tone K. A slacks-based measure of efficiency in data envelopment analysis[J]. European Journal of Operational Research, 2001, 130(3):498-509.

Comment 8:

Page 8: “In this paper, we use the DID mode”. I don’t understand what the point was to show the previous two models. I’d recommend the authors delete the discussion of the first two and focus more on the justification of the DID model if this was the one that the authors had used. Or, if the first two models explain the logic of selecting the third one, this should be better clarified.

Response 8:

Thank you for your valuable and thoughtful comments. We apologize for the confusion that our unclear expression has caused. The logic of selecting these model is, first, we use Equation (1) and Equation (2) jointly to calculate the indicator of green total factor productivity (GFTP); second, after obtaining the indicator, we exploit the DID method to examine the impact of the pilot FTZs on GTFP.

Comment 9:

Page 7, Table 1 should be placed in ‘3.3. Variables and data description’.

Response 9:

Thanks for your kind suggestions, which is valuable for improving the accuracy of the manuscript. We have placed Table 1 in ‘3.3. Variables and data description’.

Comment 10:

Parallel trend test and dynamic test” – further clarification of the used formula should be placed in Methodology. Authors should focus on Empirical results only.

Response 10:

Thanks for your constructive suggestion, which is highly appreciated. We have placed the further clarification of the used formula in Methodology, and we only focus on the empirical results in “Parallel trend test and dynamic test” section.

Comment 11:

Page 10. Figure 2 should be Figure 3. The authors need to look at the numbering of the figures throughout the paper again.

Response 11:

We would like to thank the reviewer again for your kind suggestion, we’re sorry that our carelessness has caused you inconvenience in reading the previous manuscript. We have revised and updated the order of figures in the paper.

Comment 12:

This paper doesn’t have a Discussion section. It either should be placed within the Findings chapter, or a new section should be created. In this section, authors need to clarify how their findings contrast or supplement the existing studies – those that they provided in their Literature Review. Hence, this Discussion should contain relevant citations.

Response 12:

We thank all the reviewers for helpful comments. We have added a discussion section, the specific content is as follows:

The present research contributes to the literature on policy zones which are devoted to promote FDI and international trade, and environment. Moreover, on the basis of previous research on the effect of FTZs on economic growth (Tan et al.,2015; Zhang et al., 2018) and environmental pollution (Wang ,2017), this paper further show that the urban economic growth brought by the FTZ is not at the expense of the environment. Using DID and PSM-DID, our findings are consistent with the research conclusion of Jiang et al (2021) based on the case of Shanghai FTZ. It suggests that the concept of green development applied by the central and local governments in the pilot FTZs is effective, which is not just limited to Shanghai, and the FTZs as the green zone drives the green development of cities.

Although few literatures, such as Jiang et al. (2021), examined the impact of Shanghai FTZs on green development, they did not study its underlying mechanism. The present study fills the gap and finds that the pilot FTZs promotes the green development of cities by promoting FDI and industrial structure upgrading. First, the pilot FTZs have implemented a series of institutional innovations that improve investment liberalization and facilitation, which has greatly encouraged the entry of foreign capital. It is worth noting that the pilot FTZs have strict environmental requirements for foreign investment, therefore, it mainly attracts foreign investment with high-tech, high value-added, and environmentally friendly characteristics, which is conducive to the green development of the city. In addition, the government attaches great importance to industrial planning and design in the pilot FTZs, and is committed to prioritizing the development of high-tech industry, high-end service industry, and financial services industry, thus promoting the industrial upgrading and green development.

Reference

Jiang Y, Wang H, Liu Z. The impact of the free trade zone on green total factor productivity ——evidence from the shanghai pilot free trade zone[J]. Energy Policy, 2021, (148).

Tan, N, Zhou, X, Lin, J. Study on Economic Growth Effect of Shanghai Pilot Free Trade Zone Based on Counterfactual Analysis with Panel Data[J]. Journal of International Trade, 2015(10):14-24.

Wang Y. Environmental Protection Review in the Free Trade Zone[J]. CHINESE ECONOMY,2017,50(4): 291-296.

Zhang, J,Yan, D, Feng Z, Li C. Can free trade zone effectively promote economic growth? a DID model dynamic analysis perspective[J], Inquiry into Economic Issues, 2018(11):125-133.

Comment 13:

The conclusion is quite generic. The authors should start with the initial aim of the paper and then go through their hypotheses and explain what they have found in their study according to them. Recommendations for future research can be developed further: the phrase “future studies can be further conducted from the perspective of other channels” is unclear.

Response 13:

We thank for the reviewer’s careful consideration. We have re-written the “conclusion” section, the specific content is as follows:

To promote all-dimensional, multi-layered and wide-ranging opening-up to the outside world, China set up pilot FTZs after the establishment of SEZs, EPZs, BZs and so on. Pilot FTZs not only facilitate trade and investment liberalization and facilitation through institution innovation but also take the green development as the basic concepts in the development of construction plans. For this reason, the paper devotes to investigate the impact of pilot FTZs on green and healthy development of cities using the DID and PSM-DID method. Based on theory analysis, we find that pilot FTZs have strict environmental requirements for foreign investment. Specially, pilot FTZs mainly encourage foreign investment with high-tech, high value-added, and environmentally friendly characteristics, which is benefit to the green growth of city. In addition, industry planning and design set for FTZs tend to prioritize the development of high-tech industry, high-end service industry and financial services industry, which promotes industrial upgrading and further boosts the green development of cities. Consequently, we propose the hypothesis that pilot FTZs promote the green development through FDI and industrial upgrading and confirm the mechanism by the mediation model.

The evidences indicate that, first, the pilot FTZs have significantly improved the GTFP of cities, and the results are robust to the estimation conclusion of PSM-DID. Second, pilot FTZs have a positive impact on urban GTFP by expanding the scale of FDI and promoting industrial structure upgrading. Third, there are heterogeneous effect by region and city size. The impact of pilot FTZs on GTFP is significant in central and eastern cities of China but not in the western regions. In addition, it shows a positive effect in large- and medium-sized cities, while it is not significant in small cities.

The research conclusions of this article suggest that, pilot FTZs carry out trade and investment following internationally accepted green trade rules, commercial and ecological environmental management rules. The concept of green development is penetrated to the whole process of the pilot FTZs construction. And the pilot FTZs vigorously develops modern green service industry, green manufacturing, and green supply chain, which greatly promotes the green and healthy development of the city. Indeed, the FTZs aims to develop into a green area, which has become a feature that distinguishes it from Chinese policy zones such as SEZs, EPZs, and BZs and FTZs in other countries.

European countries are at the forefront in the fields of green energy, low-carbon technology, and environmental governance technologies. China urgently needs the EU’s advanced technical support and experience sharing, and takes advantage of huge potential in clean energy market and low manufacturing cost. At present, some FTZs such as Qingdao FTZ have cooperated with Germany and other countries on sustainable production. In the future, China’s FTZs can further relax market access for EU companies in the above fields and attract them to enter China’s environmental protection market. More importantly, FTZs should actively make use of the cooperation with European countries in these fields to boost the green development of cities.

The academic research in the future should pay attention to how the pilot FTZs can stick to the concept of green development, and continue to transform itself into green areas in the future. Due to the limitation of data, this paper only can examine the short-term effect of the FTZ on urban green development. With more data available, the following researchers will have the opportunity explore the long-term effect of the pilot FTZs on urban green development. Moreover, evaluating the development status of green finance, environmentally conscious manufacturing, environmentally supply chain, etc. in FTZs is also an important research topic in the future.

Finally, we also tried our best to revised some details to improve accuracy and fluency of this paper. These changes will not influence the content and framework of the paper. And here we did not list the changes but marked them up by using the “Track Changes” function and marked them in red in the revised paper.

In all, I found the reviewer’s comments are quite helpful, and I revised my paper point-by-point. Thank you and the review again for your help!

Reviewer 2 Report

Please see attached report.

Author Response

Dear reviewer,

Thank you for your letter and for the reviewer’ comments concerning our manuscript entitled “Have China’s pilot free trade zones improved green total factor productivity? Fresh evidence from a quasi-experimental approach” (ijerph-1392324). Those comments are all valuable and very helpful for revising and improving our paper.

Reviewer:

The paper deals with a highly relevant topic, especially in the age of man-made climate change and its management, by looking at what impact free trade zones can have on resource-conserving and thus sustainable production.

The writing style of the paper is very appealing. I have a few comments regarding both the content and the form, but all of them can be implemented quickly, so I can support the publication of the paper in principle.

Response:

We thank the reviewer for acknowledging the strong performance of this work and sincerely thank for your careful and helpful consideration. According to your suggestions, we have studied comments carefully and have made correction which we hope meet with approval, the responds to your comments are as following:

Comments to the Author

Comment 1:

Equation 1 on page 6 should be explained more clearly. It is not clear to me why the expression should be minimized given that a value of 0 means complete inefficiency while the highest possible value, 1, means complete efficiency. Either it should therefore be maximized or the authors should explain why there is minimization here.

Response 1:

Thanks for your valuable suggestion! We apologize for the confusion caused by our unclear statement. The construction of formula (1) is based on the research of Tone (2001). The formula (1) cannot directly calculate the GTFP, but is used to calculate the ratio of the average input reduction and output increase. This model resolves the influence of slack variables on the measured value by incorporating the slacks of input and output factors directly into the objective function, and it effectively improves the accuracy of the measured values (Pongpanich et al., 2017; Huang et al., 2018).

The reason why the SBM model adopts the minimized form is that the projection point of the evaluated unit is the point farthest from the evaluated unit on the frontier, which is to maximize the inefficiency of input and output. In the original manuscript of Tone (2001), it is basically the same as the formula we have constructed. The following figure shows the SBM model set by Tone (2001):

Source from: Tone K. A slacks-based measure of efficiency in data envelopment analysis[J]. European Journal of Operational Research, 2001, 130(3):498-509.

However, the disadvantage of the SBM model, as you said, is that it uses a minimized form. Tone (2010) tried to design a method to calculate the shortest distance to the frontier, but it was only suitable for the case where there are fewer units to be evaluated. Although the SBM model has some shortcomings in the setting of the formula form, it is still the mainstream method of efficiency measurement due to its consideration of slack variables and is widely adopted by related literature (Pongpanich et al., 2017; Huang et al., 2018; Xie et al., 2021). Therefore, and in line with the above research, the SBM model, proposed in this paper, is con-structed as follows:

Reference

Huang Q, Hu J, Chen X. Environmental regulation, and green total factor productivity: dilemma or win-win? [J]. China Population, Resources and Environment, 2018, 28(11):140-149.

Pongpanich, R, Peng, et al. The performance measurement of listed companies of the agribusiness sector on the stock ex-change of Thailand[J]. Agricultural Economics, 2017, 63.

Tone K. A slacks-based measure of efficiency in data envelopment analysis[J]. European Journal of Operational Research, 2001, 130(3):498-509.

Xie R, Fu W, Yao S, Zhang Q. Effects of financial agglomeration on green total factor productivity in Chinese cities: Insights from an empirical spatial Durbin model[J], Energy Economics,2021,101.

Comment 2:

In the discussion of Figure 2 on page 10, it is pointed out that the effect significantly accelerates GTFP only in the two subsequent years after the establishment of FTZs, but becomes insignificant thereafter. Does the effect then only occur in the short term and is not sustainable? Is this only a short flash in the pan or a pull-forward of the development, which then falls back to its old path. Can the authors go into more detail on the causes and explanations here?

Response 2:

Thank you for your valuable and thoughtful comments. Following your suggestion, we have added more detail on the causes and explanations on Figure 3. The specific content is as follows: Figure 3 also allow us to know the dynamic effect of FTZs on GTFP. The results illustrate that the driving effect of pilot FTZs on is significant in the first and second year. It indicates that FTZs has effectively attracted the agglomeration of environment-friendly foreign capital in the zones, and supportive policies has enabled the development of some high-tech industries and green industries, which in turn promoted urban green development. However, we also find that in the third, fourth, and fifth years after the establishment of FTZs, the promotion effect is no longer significant. This may be because the imperfect policy guarantee systems, especially the failure to establish an effective intellectual property system, have interfered the positive impact of the FTZs on GTFP (Zhuo et al., 2021). And we can see that the effect of FTZs on GTFP is showing an upward trend from the third year. Due to the unavailability of data, we are not able to estimate the long-term effect of FTZs on GTFP. Since the ideas of green development are continuously been applied in FTZs, we can infer that there will exist a long-term positive effect. In the future, the governments should continue to adhere to the ideas of green development in the FTZs, therefore they can effectively drive urban green development.

Reference

Zhuo C, Mao Y, Rong J. Policy dividend or “policy trap”? Environmental welfare of establishing free trade zone in China[J]. Science of The Total Environment, 2021, 756, 143856.

Comment 3:

In principle, the models show significant results, but the explanatory power indicated by R² is very low. This should also be discussed at least briefly.

Response 3:

Thanks for your valuable suggestion. This paper uses fixed effects model for regression. All Tables in this paper control for fixed effects, so the R2 is relatively small. Previous studies have suggested that in this case, a relatively small R2 does not affect the robustness of the conclusion. For example, Lemmon et al., (2008) have illustrated that 92.45% of R2 is explained by the fixed effect. Bodt et al., (2019) have pointed that the after controlling for individual fixed effects, R2 obtained by using the least squares method is only 5%.

Indeed, it is relatively common in the existing literature to obtain a relatively small R2 when using the DID method to evaluate the economic effects of policy experiments. For example, employing a DID method, R2 obtained by the research of Malin et al., and Lu et al., (2020) are only 0.009, 0.0002, respectively. A more relevant document is Li et al., (2021), based on panel data, the scholar uses the DID method to study the impact of Pilot FTZs on Chinese port-listed companies’ performance, after controlling for individual fixed effects and time fixed effects, the R2 of Table 5 is only 0.0426.

Reference

Bodt E D, Cousin J G, R Roll. Improved method for detecting acquirer fixed effects[J]. Journal of Empirical Finance, 2019, 50:20-42.

Lemmon M L, Roberts M R, Zender J F. Back to the Beginning: Persistence and the Cross-section of Corporate Capital Structure[J]. Journal of Finance, 2008, 63(4):1575–1608.

Li S, Liu J, Kong Y. Pilot Free Trade Zones and Chinese Port-listed companies Performance: An Empirical Research Based on Quasi-Natural Experiment[J]. Transport Policy, 2021.

Lu X, Chen D, Kuang B, et al. Is high-tech zone a policy trap or a growth drive? Insights from the perspective of urban land use efficiency[J]. Land Use Policy, 2020, 95.

Malin S, et al. The impact of low-carbon city construction on ecological efficiency: Empirical evidence from quasi-natural experiments[J]. Resources, Conservation and Recycling, 157.

Comment 4:

The classification of cities on page 13 is not clear to me. Is the size determined on the basis of population
size (then the population thresholds could simply be given) or in relation to GDP (then the corresponding thresholds could also be given). Please explain this better!

Response 4:

We appreciate for your valuable comment. Following your suggestion, according to the classification of cities of the Notice of the State Council on Adjusting the Standards for Categorizing City Sizes, No. 51 [2014], State Council of China. We have redefined the large and medium-sized cities and small cities in China on the basis of the size of the urban population. Specifically, if the population of a city is more than 500,000, we define it as a large and medium-sized city, and if the population of a city is less than 500,000, we define it as a small city.

Comment 5:

It would be interesting to know which countries the cities trade with and how strong the influence is here. European countries in particular now attach great importance to sustainable production, i.e. if the Chinese regions under consideration produce for these markets, this can promote the conversion of the economy to meet the required standards. If the trading partners are not so interested in sustainable production, the conversion does not have to take place.

Response 5:

Thank you for your valuable suggestion. We expressed our interest in this topic in the conclusion of the paper, the specific content is as follows:

European countries are at the forefront in the fields of green energy, low-carbon technology, and environmental governance technologies. China urgently needs the EU’s advanced technical support and experience sharing, and takes advantage of huge potential in clean energy market and low manufacturing cost. At present, some FTZs such as Qingdao FTZ have cooperated with Germany and other countries on sustainable production. In the future, China’s FTZs can further relax market access for EU companies in the above fields and attract them to enter China’s environmental protection market. More importantly, FTZs should actively make use of the cooperation with European countries in these fields to boost the green development of cities.

The academic research in the future should pay attention to how the pilot FTZs can stick to the concept of green development, and continue to transform itself into green areas in the future. Due to the limitation of data, this paper only can examine the short-term effect of the FTZ on urban green development. With more data available, the following researchers will have the opportunity explore the long-term effect of the pilot FTZs on urban green development. Moreover, evaluating the development status of green finance, environmentally conscious manufacturing, environmentally supply chain, etc. in FTZs is also an important research topic in the future.

Minor Comments

  1. Some abbreviations are not explained in the text but are provided in the abstract. However, these should also be explained in the actual text when used for the first time (FTZ p. 1; GTFP p. 2).
    2. Fig. 1 is not necessary, since the information content has already been presented in full in the text.
    3. On p. 3 there should be a period before “Finally, Conclusions are …” and no period after “trade liberalization.; more …”
    4. Figure 2 should be revised. To improve clarity, it is advisable to display the years at an angle. In addition, the units (billion yuan) should be indicated on the axes.
    5. On p. 5, the source Melitz 2003 was cited wrong (he has no co-authors as indicated by “et al.”)
    6. On p. 6, the illustration is incorrectly numbered (this should be 3 here). In addition, the quality should be improved and the paragraph arrow (8 ) after the words should be removed.
    7. On p. 6 in 3.1.1, Tone is mentioned. However, the source is not cited in the references (or is it number 12?).
    8. On p. 7, Malmquist (also with year) is mentioned. However, the source is not cited in the references.
    9. Just to be sure, since this gas is otherwise very prominent: in Table 1 on p. 8, exhaust gas means SO2 and not CO2?
    10. On p. 9, table 2, the legend is not relevant as there is no significance in descriptive statistics.
    11. On p. 9 in the fourth last line there is a character displayed incorrectly, I suppose it should be ∀ ?? ≠ 1?
    12. On p. 10, figure 2 should be figure 4.
    13. On p. 11, figure 3 should be figure 5.

Response minor comments

We appreciate and admire your attitude very much. Although these are small problems, they have played an important role in the improvement of our article. We have carefully revised the full text according to your opinions.

  1. In the introduction, we corrected the abbreviations of pilot free trade zone and green total factor productivity.
  2. According to your suggestion, we have deleted Figure 1.
  3. We have corrected this.
  4. We revised figure 2 and added the units billion yuan (RMB) to make it more beautiful and clearer.
  5. We have corrected this.
  6. We have corrected this.
  7. We have added this reference.
  8. We have added this reference.
  9. Thanks for your kind suggestions. CO2 is indeed an important part of exhaust gas. However, since the Chinese government has not published the city-level industrial carbon dioxide emission data, in order to make the research of this paper more scientific, we have adopted the indicator (industrial SO2 emissions), which is also an important part of exhaust gas. In addition, it is explained in the form of footnotes.
  10. We have deleted this.
  11. We have revised this, which means K ≠ 1.
  12. We have corrected this.
  13. We have corrected this.

Finally, we also tried our best to revised some details to improve accuracy and fluency of this paper. These changes will not influence the content and framework of the paper. And here we did not list the changes but marked them up by using the “Track Changes” function and marked them in red in revised paper.

We appreciate for the reviewers’ warm work earnestly, and hope that the correction will meet with approval.

Once again, thank you very much for your comments and suggestions.

Round 2

Reviewer 1 Report

The authors have made significant progress from the last time. This paper has excellent potential for publication. However, more work is needed.

First of all, further English editing is needed. Mostly, polishing for typos and minor grammar mistakes instead of rewriting big parts of the text.

For example, some sentences are too long, e.g., Page 1, where the definition of FTZ was given.

Also, authors must avoid starting the sentence with “And”. This is a mistake throughout the text.

Page 2: authors provided a historical overview of the FTZ development in China, which is good. But it looks quite vague. Authors need to add some key dates and perhaps names of the key FTZ implemented in China since the 70s.

Page 2: ‘Third, introducing the negative list management model, and foreign capital is allowed to enter areas which are not in the list.’ – this is a swiping statement. What list? What does it include? A bit more clarification is needed here. Also, there’s a footnote “1”, but where is it?

Page 2: ‘In addition, Guiding Opinions on Strengthening…’ – it’s not really clear what authors are trying to say here. This paragraph came out of nowhere. Perhaps, if they include dates or a connecting statement, this would be helpful. Authors further used the word ‘should’ – so are they talking about the future recommendation or the past? Further clarification or rephrasing is needed here. Also, the sentence in this paragraph is very long – it should be divided.

Page 3: I’m not sure that the use of the word ‘remainder’ is appropriate. Simply, use ‘the structure of the paper is the following' or similar.

In the Introduction or Literature Review, the authors observed some existing studies on the topic but mostly covered ‘positive’ contributions of the SEZ on the local economy. In academic writing, critical evaluation is necessary. There are some critical studies about SEZ impact (negative) on job employment, environment, property aspect, tax returns, etc. A few critical sentences will be useful.

Also, the part ‘Institutional background’ is merely descriptive. Similarly, authors need to include a couple of sentences about the institutional impact on SEZ development: e.g., government impact or intervention. Some examples:

Sosnovskikh, S., & Cronin, B. (2021). The effects of culture, attitudes and perceptions on industrial cluster policy: The case of Russia. Competition & Change, 25(3–4), 350–381. https://doi.org/10.1177/1024529420949491

Aritenang, A. F., & Chandramidi, A. N. (2019). The Impact of Special Economic Zones and Government Intervention on Firm Productivity: The Case of Batam, Indonesia. Bulletin of Indonesian Economic Studies, 1–40. https://doi.org/10.1080/00074918.2019.1643005

Vernay, A.-L., D’Ippolito, B., & Pinkse, J. (2018). Can the government create a vibrant cluster? Understanding the impact of cluster policy on the development of a cluster. Entrepreneurship & Regional Development, 30(7–8), 901–919. https://doi.org/10.1080/08985626.2018.1501611

Page 7: ‘undesired output’ – I’m not sure what it means. Please, clarify or eliminate.

Page 7: a citation is needed for ‘The traditional data envelopment analysis model’.

Page 10: ‘Since the Chinese offcial government has not publish the city-level CO2 industrial emissions data, the industrial SO2 emissions is adopted here.’ – there should be ‘official’ and ‘published’. Generally, as I mentioned above, the text needs further proof-reading for typos and minor grammar mistakes.

Also, I don’t have any questions about the Findings chapter. I can see that authors are excellent at analysing data. But, writing and narrating skills should be improved further. The authors provided some links to existing studies while discussing the findings; however, there are still paragraphs that are not cited, which is unprofessional. I can clearly see that when they explain the findings, certain statements are not supported. They can be supported by whatever was even discussed and observed in the Literature Review. For example, on Page 15: ‘In general, compared with smaller cities, larger cities tend to have advantages…’ – in this paragraph, the authors discussed their findings and provided some explanations. To avoid presenting this as a ‘personal discussion’ and ‘assumptions’, citations are needed.

The conclusions are good, informative and clear; but, referencing is also needed.

Author Response

Dear reviewer,

Thank you again for your letter and for the comments concerning our manuscript entitled “Have China’s pilot free trade zones improved green total factor productivity? Fresh evidence from a quasi-experimental approach” (ijerph-1392324). Those comments are all valuable and very helpful for revising and improving our paper.

Reviewer:

The authors have made significant progress from the last time. This paper has excellent potential for publication. However, more work is needed.

#Round2 Response:

We thank the reviewer for acknowledging the strong performance of the work we have done in the first round of revision, and sincerely thank again for your careful and helpful consideration. According to your suggestions, we have studied comments carefully and have made correction which we hope meet with approval, the responds to your comments are as following:

Comments to the Author

Comment 1:

First of all, further English editing is needed. Mostly, polishing for typos and minor grammar mistakes instead of rewriting big parts of the text.

Response 1:

Thank you for your valuable and thoughtful comments. We have carefully checked and improved the English writing in the revised manuscript. In addition, we have consulted a professor, who’s a native English expert, to polish our paper. Please see if the revised version met the English presentation standard.

Comment 2:

For example, some sentences are too long, e.g., Page 1, where the definition of FTZ was given.

Response 2:

Thank you for your valuable and thoughtful comments. Following your suggestion, we find that more refined sentences will indeed make our article more accurate and more appropriate, so we have revised some long sentences.

Sentence 1: “According to Teifenbrun (2015) and Akbari et al. (2019), FTZ refers to an area where goods may be landed, handled, manufactured, or reconfigured, and then re-exported without the intervention of customs authorities, and multiple institutional incentives such as free movement of goods and people, and preferential taxation are provided in it.”

Revised sentence 1: “According to Teifenbrun (2015) and Akbari et al. (2019), FTZ refers to an area where goods may be landed, handled, manufactured, or reconfigured, and then re-exported without the intervention of customs authorities. In the zones, multiple institutional incentives such as free movement of goods and people, and preferential taxation are provided.”

Sentence 2: “Regarding FTZs, it is generally believed that Chinese FTZs have reduced environmental pollution, because in the FTZs, various measures such as clean production mechanisms are taken by the government to control pollution.”

Revised sentence 2: “Regarding FTZs, it is generally believed that Chinese pilot FTZs have reduced environmental pollution, since various measures such as clean production mechanisms are adopted by the government for pollution prevention and control.”

Sentence 3: “GTFP is a new productivity indicator that comprehensively considers resource and environmental constraints, specifically, the new indicator adds resource consumption as an input factor into the traditional TFP analysis framework and adds environmental pollution emissions as an undesirable output into the input-output efficiency analysis (Jefferson et al., 2008; Jones et al., 2011; Zhu et al., 2018), which can effectively reflect the level of green and healthy development (Tian and Lin, 2017, Liu and Xin, 2019).”

Revised sentence 3: “GTFP is a new productivity indicator that comprehensively incorporates resource and environmental constraints. Specifically, the new indicator adds resource consumption as an input factor to the traditional TFP analysis framework and adds environmental pollution emissions as an undesirable output to the input-output efficiency analysis (Jefferson et al., 2008; Jones et al., 2011; Zhu et al., 2018), which can effectively reflect the level of green and healthy development (Tian and Lin, 2017, Liu and Xin, 2019).”

Sentence 4: “However, concerning China, regarding pilot FTZs and green development, only few researchers like Jiang et al. (2021) conducted research on Shanghai FTZ and believed that FTZ is a great incentive to promoted green development, while Zhuo et al. (2021) took the Guangdong FTZ as the research object, concluded that FTZ is at the expense of the environment, for every 100 million yuan increase in the GDP, discharged wastewater and waste gas will increase by 1.746 million tons and 28.016 tons, respectively.”

Revised sentence 4: “However, concerning China, regarding pilot FTZs and green development, only few researchers like Jiang et al. (2021) conducted research on Shanghai FTZ and believed that FTZ is a great incentive to promoted green development. However, Zhuo et al. (2021) took the Guangdong FTZ as the research object and concluded that this FTZ operates at the expense of the environment, for every 100 million yuan increase in the GDP, discharged wastewater and waste gas will increase by 1.746 million tons and 28.016 tons, respectively.”

Sentence 5: “Shenzhen is committed to ensuring 100% coverage of green buildings and establishing a green and creative transportation system and central cooling project in the zone to reduce pollutant emissions.”

Revised sentence 5: “Shenzhen is committed to ensuring 100% coverage of green buildings, establishing a green and creative transportation system, and constructing a central cooling project in the zone to reduce pollutant emissions.”

Sentence 6: “The continuous flow of advanced production factors to the industry with high added value can not only facilitate the development of import and export trade and service trade and enhance the competitiveness of related industries but also optimize the industrial structure and promote industrial upgrading.”

Revised sentence 6: “Advanced production factors continuous flow to high value-added industries, which can not only facilitate the development of import, export, and service trade, enhance the competitiveness of related industries, but also optimize the industrial structure and promote industrial upgrading.”

Etc.

We have made language and structure changes to more sentences; these details can be reviewed in the revised manuscript.

Comment 3:

Also, authors must avoid starting the sentence with “And”. This is a mistake throughout the text.

Response 3:

Thanks for your valuable suggestion. We have corrected these mistakes throughout the text. Specifically, we replaced them with other conjunctions.

Comment 4:

Page 2: authors provided a historical overview of the FTZ development in China, which is good. But it looks quite vague. Authors need to add some key dates and perhaps names of the key FTZ implemented in China since the 70s.

Response 4:

We appreciate for your valuable comment. According to your suggestion, and in order to make the historical overview of the FTZ development look clearer, first, we have added some key dates and names of FTZs in here, the specific content is as follows: “The Chinese government has always been committed to deepening reform and opening wider to the outside world. In 1980, Shenzhen, Zhuhai, Shantou, and Xiamen have been approved as special economic zone (SEZ). Then, China established the first bond-ed zone (BZ) in Shanghai in 1990, and successively set up BZs in Tianjin, Dalian, and other cities. In 2000, China approved 15 pilot export processing zones (EPZs) including Yantai Weihai, Hangzhou, etc. at one time.  Since then, China continuously established some other SEZs, BZs, and EPZs. On Sept. 29, 2013, China’s first FTZ— Shanghai pilot FTZ was established, which marks a new stage of China’s opening-up. After that, more pilot FTZs were built in Tianjin, Chongqing, and other cities by March 2017.”

In addition, we have added a footnote here to introduce the names of free trade zones in China and when they were established. The specific content of footnote 2 is “Up to now, China has established six batches of FTZs: the first batch includes Shanghai FTZ (2013); the second batch include: Guangzhou (2015), Shenzhen (2015), Zhuhai (2015), Tianjin (2015), Fuzhou (2015), Xiamen (2015); the third batch include: Shenyang (2017), Yingkou (2017), Dalian (2017), Zhoushan (2017), Kaifeng (2017), Luoyang (2017), Zhengzhou (2017) Yichang (2017), Wuhan (2017), Xiangyang (2017), Chongqing (2017), Chengdu (2017), Luzhou (2017), Xi’an (2017). After that, the Chinese government successively established the fourth, fifth and sixth batch of free trade zones in different cities. Due to data limitations, this paper considers only the FTZs in first, second batches.”

Comment 5:

Page 2: ‘Third, introducing the negative list management model, and foreign capital is allowed to enter areas which are not in the list.’ – this is a swiping statement. What list? What does it include? A bit more clarification is needed here. Also, there’s a footnote “1”, but where is it?

Response 5:

Thanks very much for your comment, which is highly appreciated. Considering the reviewer’s suggestion, we have added a footnote to illustrate the term “negative list”. Specific description is listed as follows: “The Negative List provides guidance and governs industry sectors in which foreign investment is prohibited or where possible restrictions may apply (Zhang and Xiang, 2013; Wang, 2014). The Special Administrative Measures (Negative List) on Foreign Investment Access into the China (Shanghai) Free Trade Pilot Zone (2013) stated that 18 industries including news organizations, the publishing businesses of books, newspapers, and periodicals, and the gambling industry, etc., are all “forbidden areas” for foreign investment.”

In addition, we’re sorry that our carelessness has caused you inconvenience in reading the previous manuscript. The position of footnote 1 is after the sentences: “The institutional innovations mainly include, first, optimizing government functions, and improving trade and investment efficiency through one-window service, etc. Second, issuing tax holidays and tax regimes such as reduced income tax for some enterprises in FTZs. Third, introducing the negative list management model, and foreign capital is allowed to enter in areas that are not on the list.” The footnote is used to list the relevant policies referred to in this passage. Moreover, it is also worth noting that it is now footnote 4.

Reference

Zhang, X, Xiang, P. Negative List: New Challenges of China’s Opening-up[J]. Intertrade, 2013(11):19-22.

Wang, Z. An International Comparison of Transformation Experiences of “Negative List” and its Enlightenment to China[J]. International Economics and Trade Research,30(9):72-84.

Comment 6:

Page 2: ‘In addition, Guiding Opinions on Strengthening…’ – it’s not really clear what authors are trying to say here. This paragraph came out of nowhere. Perhaps, if they include dates or a connecting statement, this would be helpful. Authors further used the word ‘should’ – so are they talking about the future recommendation or the past? Further clarification or rephrasing is needed here. Also, the sentence in this paragraph is very long – it should be divided.

Response 6:

Thanks for your kind suggestions, which is valuable for improving the accuracy of the manuscript. Following your suggestions, first, we have added date and connecting statements to this paragraph. Second, after clarifying when the policy was issued, we can see that the word ‘should’ mean that in the future, the ideas of green development should be penetrated to the whole process of the FTZs construction. Third, we have divided the sentences in this paragraph to make them short and accurate. The revised paragraph is as follows:

“Green development” is a major development ideas of Chinese pilot FTZs. As the concepts emphasized in the “Guiding Opinions on Strengthening Ecological Environment Protection in Pilot Free Trade Zones”, the ideas of green development should penetrate to the entire process of pilot FTZs construction, and pilot FTZs must develop a modern green service industry, green manufacturing, green supply chain, and green trade in the future. Some studies also indicated that pilot FTZs make efforts to become green areas in various ways (An and Zi 2020). Therefore, as green areas, have the pilot FTZs effectively promoted urban green development? This paper attempts to address this question and provide useful policy alternatives.”

Reference

An, G, Zi, W. Research on Green Finance in the Aspect of Promoting the Sustainable Development of the Free Trade Zone[J]. Public Finance Research, 2020(5):117-129.

Comment 7:

Page 3: I’m not sure that the use of the word ‘remainder’ is appropriate. Simply, use ‘the structure of the paper is the following’ or similar.

Response 7:

Thank you very much for your recommendation. We have replaced ‘remainder’ with “The structure of the rest of paper is as follows”.

Comment 8:

In the Introduction or Literature Review, the authors observed some existing studies on the topic but mostly covered ‘positive’ contributions of the SEZ on the local economy. In academic writing, critical evaluation is necessary. There are some critical studies about SEZ impact (negative) on job employment, environment, property aspect, tax returns, etc. A few critical sentences will be useful.

Response 8:

Thanks very much for your comments, which are very helpful to improve the quality of this article. It is really true that some critical studies about SEZ impact is useful to improve our paper. Therefore, according to your suggestion, we have added some literature on negative opinions. The specific content is as follows: “Although few studies have attempted to estimate the impact of policy zones on urban green and healthy development, numerous literatures have explored their im-pact on economic growth. Most extant studies support that SEZs, EPZs, and BZs positively contribute to innovation stimulation and economic growth (Wu et al., 2020; Ye, 2020). However, additional critical findings suggest that these policies also have some dampening effects such as resource mismatches (Zhou, 2017), widening the gap be-tween the rich and poor (John, 1998), and tax evasion (Julien and Ji, 2020), etc. Concerning FTZs, the existing literature mainly focuses on positive effects…”

Reference

Julien, Chaisse, Ji, X. The Pervasive Problem of Special Economic Zones for International Economic Law: Tax, Investment, and Trade Issues[J]. World Trade Review, 2020, 19.

John, M, Litwack, et al. Balanced or Unbalanced Development: Special Economic Zones as Catalysts for Transition[J]. Journal of Comparative Economics, 1998.

Zhou, J. The negative impact of China’s regional economic differences[J]. Review of Economic Research, 2017(6):33.

Comment 9:

Also, the part ‘Institutional background’ is merely descriptive. Similarly, authors need to include a couple of sentences about the institutional impact on SEZ development: e.g., government impact or intervention. Some examples:

Sosnovskikh, S., & Cronin, B. (2021). The effects of culture, attitudes, and perceptions on industrial cluster policy: The case of Russia. Competition & Change, 25(3–4), 350–381. https://doi.org/10.1177/1024529420949491

Aritenang, A. F., & Chandramidi, A. N. (2019). The Impact of Special Economic Zones and Government Intervention on Firm Productivity: The Case of Batam, Indonesia. Bulletin of Indonesian Economic Studies, 1–40. https://doi.org/10.1080/00074918.2019.1643005

Vernay, A.-L., D’Ippolito, B., & Pinkse, J. (2018). Can the government create a vibrant cluster? Understanding the impact of cluster policy on the development of a cluster. Entrepreneurship & Regional Development, 30(7–8), 901–919. https://doi.org/10.1080/08985626.2018.1501611

Response 9:

Thank you for pointing this problem out and providing additional literature for reference. We have carefully read the article you listed and added a couple of sentences about the institutional impact on SEZs, EPZs, BZs etc. development. The specific content is as follows: “Since 1978, China has established SEZs, the zones were designed as a major platform that provided preferential policies such as tax reduction for foreign enterprises. In addition, BZs were successfully constructed since the 1990s, goods are exempt from duty in the zones. Further, the Chinese government implemented the EPZs policy including goods that are manufactured for export will be exempt from tax.

Government’s peculiarities and intervention plays an important role in the development of these policy zones (Vernay et al., 2018; Aritenang and Chandramidi, 2019; Sosnovskikh and Cronin, 2021). In the Chinese context, the key experiences of SEZs, EPZs, BZs etc. can best be summarized as gradualism with an experimental approach; a strong commitment; and the active, pragmatic facilitation of the state (Zeng, 2011). Specifically, first, the Chinese government provides preferential policies and broad institutional autonomy such as duty-free benefits, which greatly promoted industrial clusters. Second, strong support and proactive participation of governments, especially in the areas of public goods and externalities. It ensured a stable and supportive envi-ronment for long-term development. Third, the government continuous promotes technology learning and upgrading by designing policies and foresight activities.

Under the positive impact and intervention of government, most of special zones in China, though differing in characteristics, are quite successful in FDI introduction and trade prosperity. However, there is still considerable room for China to expand the opening of its service industry and attract advanced foreign investment.”

Reference

Sosnovskikh, S., & Cronin, B. (2021). The effects of culture, attitudes, and perceptions on industrial cluster policy: The case of Russia[J]. Competition & Change, 25(3–4), 350–381.

Aritenang, A. F., & Chandramidi, A. N. (2019). The Impact of Special Economic Zones and Government Intervention on Firm Productivity: The Case of Batam, Indonesia[J]. Bulletin of Indonesian Economic Studies, 1–40.

Vernay, A.-L., D’Ippolito, B., & Pinkse, J. (2018). Can the government create a vibrant cluster? Understanding the impact of cluster policy on the development of a cluster[J]. Entrepreneurship & Regional Development, 30(7–8), 901–919.

Zeng, D, Z. How do special economic zones and industrial clusters drive China's rapid development? [J]. Policy Research Working Paper, 2011:1-53.

Comment 10:

‘undesired output’ – I’m not sure what it means. Please, clarify or eliminate.

Response 10:

Thanks for your suggestion, we are sorry that our carelessness has caused your inconvenience in reading. In the previous manuscript, in Page 8, we wrote x, y, and b represent the vectors of the input factors, expected outputs and undesired outputs. After serious consideration of your suggestions, we refer to the relevant classic literature like Chung and Fare (1997) and change the original term “expected outputs” to “desirable outputs” as well as changing the original term “undesired outputs” to “undesirable outputs”. And the usage is unified in the full text.

In addition, referring to the research of Chung and Fare (1997) and Rolf et al. (2005), “desirable outputs” is that strong (good) outputs, such as paper or electricity; “undesirable outputs” is that weak (bad) outputs, such as suspended solids or SO2. For the above description, we also put it in the revised manuscript in the form of footnotes. The specific content is as follows: “Desirable outputs is that strong (good) outputs, such as paper or electricity etc.; Undesirable outputs is that weak (bad) outputs, such as suspended solids or SO2 etc. (Chung and Fare, 1997; Rolf et al., 2005). In this paper, to measure desirable outputs, we observe two factors including urban output and resident quality of life. The undesirable outputs is jointly measured by four bad factors, which include wastewater, exhaust gas, soot, and haze pollution.”

Reference

Chung Y, Fare R. Productivity and Undesirable Outputs: A Directional Distance Function Approach[J]. Microeconomics, 1997, 51(3):229-240.

Rolf, Färe, and, et al. Characteristics of a polluting technology: theory and practice[J]. Journal of Econometrics, 2005.

Comment 11:

Page 7: a citation is needed for ‘The traditional data envelopment analysis model’.

Response 11:

Thank you for your helpful suggestion, we have added citations to the paragraph of the DEA model. The specific content is as follows: “The traditional data envelopment analysis (DEA) model has advantages in managing the problems of multi-input indices and multi-output indices (Hu et al., 2006; Cooper et al., 2007); thus, it is widely used in the field of development performance measurement. However, the traditional DEA method only considers economic benefits when measuring efficiency values but does not consider the impact of undesirable output. This fact is not only inconsistent with the actual production process but also ignores the slack problem of input and output factors; accordingly, there is a deviation between the measured value and the actual value (Lawrence et al., 2002).”

Reference

Hu, J, L, Wang S C. Total-factor energy efficiency of regions in China[J]. Energy Policy, 2006, 34(17):3206-3217.

Cooper, W, Seiford, L, M, Tone, K. Data Envelopment Analysis: A Comprehensive Text with Models, Applications, References and DEA-Solver Software[M]. Kluwer Academic Publishers, 2007.

Lawrence, M. Seiford, Joe, Zhu. Modeling undesirable factors in efficiency evaluation[J]. European Journal of Operational Research, 2002, 142(1):16-20.

Comment 12:

Page 10: ‘Since the Chinese official government has not publish the city-level CO2 industrial emissions data, the industrial SO2 emissions is adopted here.’ – there should be ‘official’ and ‘published’. Generally, as I mentioned above, the text needs further proof-reading for typos and minor grammar mistakes.

Response 12:

Thanks for your careful comments, I have corrected the grammar mistakes. In addition, we have carefully revised the manuscript according to the reviewers’ comments, and have re-scrutinized to improve the English writing by a language polishing service.

Comment 13:

Also, I don’t have any questions about the Findings chapter. I can see that authors are excellent at analyzing data. But writing and narrating skills should be improved further. The authors provided some links to existing studies while discussing the findings; however, there are still paragraphs that are not cited, which is unprofessional. I can clearly see that when they explain the findings, certain statements are not supported. They can be supported by whatever was even discussed and observed in the Literature Review. For example, on Page 15: ‘In general, compared with smaller cities, larger cities tend to have advantages…’ – in this paragraph, the authors discussed their findings and provided some explanations. To avoid presenting this as a ‘personal discussion’ and ‘assumptions’, citations are needed.

Response 13:

I quite appreciate your favorite consideration and the reviewer’s insightful comments. First, concerning writing and narrating skills, we have asked a professor, who is a native English speaker, to polish our paper, then, and we have re-scrutinized to improve the English writing by a language polishing service. Second, it is really true that citations are needed to make certain statements professional. Therefore, we have added citations to the paragraphs and sentences that discussed our findings and provided some explanations. We list the sentences that have been added newly references as follows:

Sentence 1: “Urban GTFP can accurately describe and reflect the development efficiency of urban economies under the constraints of resources and the environment (Fare et al., 2010).”

Reference

Fare, R., Grosskopf, S. & Pasurka, C.A. Accounting for Air Pollution Emissions in Measures of State Manufacturing Produc-tivity Growth [J]. Journal of Regional Science, 2010, 41, 381-409.

Sentence 2: “Referring to Liu and Zhao (2015), this paper further uses the propensity score matching Differences-in-Differences (PSM-DID) approach to test the robustness of the above results.”

Reference

Liu, R, Zhao, R. Western Development: Growth Drive or Policy Trap --An Analysis Based on PSM-DID Method[J]. China Industrial Economics, 2015(6):32-43.

Sentence 3: “There are numerous differences in the level of economic development, marketization, and institutional quality among the various regions in China (Wang and Dong, 2018).”

Reference

Wang, Q, Dong, Y. Research on the Regional Difference and Distributional Dynamic Evolution of Real Economy Development[J]. The Journal of Quantitative & Technical Economics, 2018, 35(5):77-94.

Sentence 4: “Following the research of Gong and Shen (2011). We divided the samples into three categories: the eastern, central, and western regions to explore the heterogeneous effects of FTZ policy on GTFP.”

Reference

Gong, J, Shen, K. Regional Distribution and Environment Pollution of Energy- intensive Industries in China [J]. The Journal of Quantitative & Technical Economics, 2011, 28(2):20-36.

Sentence 5: “In general, compared with smaller cities, larger cities tend to have advantages in industrial structure, resource agglomeration, and scientific and technological development (Lu and Chen, 2004; Pflüger and Tabuchi, 2019), which may render the impact of pilot FTZs on GTFP heterogeneous.”

Reference

Lu, M, Chen, Z. Urbanization, Urban-Biased Economic Policies and Urban-Rural Inequality[J]. ECONOMIC RESEARCH JOURNAL, 2004, 39(6):50-58.

Pflüger, M, Tabuchi, T. Comparative Advantage, Agglomeration Economies and Trade Costs[J]. Journal of Urban Economics, 2019, 109, 1-13.

Comment 14:

The conclusions are good, informative and clear; but, referencing is also needed.

Response 14:

Thanks for your valuable suggestion, we have added citations to the paragraphs and sentences in the “Conclusion” section that discussed our findings and provided some explanations.

Finally, we also tried our best to revised some details to improve accuracy and fluency of this paper. These changes will not influence the content and framework of the paper. And here we did not list the changes but marked them up by using the “Track Changes” function in revised paper.

We appreciate for the reviewers’ warm work earnestly, and hope that the correction will meet with approval.

Once again, thank you very much for your comments and suggestions.
